# It Takes Two to Tango, Part II: Synthesis of A-Ring Functionalised Quinones Containing Two Redox-Active Centres with Antitumour Activities

**DOI:** 10.3390/molecules28052222

**Published:** 2023-02-27

**Authors:** Joyce C. Oliveira, Renato L. de Carvalho, Hugo G. S. Sampaio, João Honorato, Javier A. Ellena, Felipe T. Martins, João V. M. Pereira, Pedro M. S. Costa, Claudia Pessoa, Rafaela S. Ferreira, Maria H. Araújo, Claus Jacob, Eufrânio N. da Silva Júnior

**Affiliations:** 1Institute of Exact Sciences, Department of Chemistry, Universidade Federal de Minas Gerais, UFMG, Belo Horizonte 31270-901, Brazil; 2São Carlos Institute of Physics, Physics and Interdisciplinary Sciences Department, Universidade de São Paulo, USP, São Carlos 13560-970, Brazil; 3Chemistry Institute, Universidade Federal de Goiás, UFG, Goiânia 74690-900, Brazil; 4Department of Physiology and Pharmacology, Universidade Federal de Ceará, UFC, Fortaleza 60430-270, Brazil; 5Biological Sciences Institute, Biochemistry and Immunology Department, Universidade Federal de Minas Gerais, UFMG, Belo Horizonte 31270-901, Brazil; 6Division of Bioorganic Chemistry, School of Pharmacy, University of Saarland, 66123 Saarbruecken, Germany

**Keywords:** click chemistry, triazoles, quinones, redox centres, anticancer activity

## Abstract

In 2021, our research group published the prominent anticancer activity achieved through the successful combination of two redox centres (*ortho*-quinone/*para*-quinone or quinone/selenium-containing triazole) through a copper-catalyzed azide-alkyne cycloaddition (CuAAC) reaction. The combination of two naphthoquinoidal substrates towards a synergetic product was indicated, but not fully explored. Herein, we report the synthesis of 15 new quinone-based derivatives prepared from click chemistry reactions and their subsequent evaluation against nine cancer cell lines and the murine fibroblast line L929. Our strategy was based on the modification of the A-ring of *para*-naphthoquinones and subsequent conjugation with different *ortho*-quinoidal moieties. As anticipated, our study identified several compounds with IC_50_ values below 0.5 µM in tumour cell lines. Some of the compounds described here also exhibited an excellent selectivity index and low cytotoxicity on L929, the control cell line. The antitumour evaluation of the compounds separately and in their conjugated form proved that the activity is strongly enhanced in the derivatives containing two redox centres. Thus, our study confirms the efficiency of using A-ring functionalized *para*-quinones coupled with *ortho*-quinones to obtain a diverse range of two redox centre compounds with potential applications against cancer cell lines. Here as well, it literally takes two for an efficient tango!

## 1. Introduction

Cancer has become a global issue and represents nearly one in six worldwide annual deaths, according to the World Health Organization [1]. Different therapies are available nowadays for many types of cancer; however, the drugs currently applied commonly lead to painful side effects, in general due to the absence of a high degree of selectivity between a cancer cell on the one side and a healthy cell on the other [2,3]. In this context, the development and subsequential evaluation of new potential anticancer compounds have been explored extensively throughout the years [4,5,6,7]. From this perspective, important bioactive molecules with prominent antitumour activity have been described [8,9,10]. Amongst these molecules, quinones in general play an important role [11,12,13,14], since they actively participate in the molecular stress generated by reactive oxygen species (ROS) [15,16], culminating in the apoptosis of the target cell. Quinones are also well-known to present valuable diverse bioactivities, including against *Trypanosoma cruzi* [17], malaria [18], *Aedes aegypti* [19], and tuberculosis [20]. Yet their special status in the fight against cancer can be highlighted due to the potent activity of substances such as *β*-lapachone [21] and respective A- [22] and C-ring [23] modified derivatives, juglone [24], menadione [25], or even more complex quinones, such as vitamin K [26] (Figure 1A).

Amongst these examples, *β*-lapachone stands out as one of the most important ones, due to its notable biological properties. It is an *ortho*-quinone derived from lapachol, and it is present in the bark of a South American tree locally known as *ipe* (*Handroanthus impetiginosus, purple ipe*) [27]. This compound is capable of leading to “programmed necroptosis” (apoptosis + necrosis) of cancer cells, caused by the increase in the concentration of ROS. In essence, cancer cells already present in a concentration of NAD(P)H:quinone oxidoreductase 1 (NQO1) are 5- to 200-fold greater than in normal cells. In these cells, *β*-lapachone triggers a redox cycle, which results in the generation of the above-mentioned ROS. The excess of ROS pushes cancer cells over a critical redox threshold. It causes a DNA single-strand breakage, overactivation of poly(ADP-ribose) polymerase-1 (PARP-1), loss of the NAD+ and ATP pools, and finally, “necroptosis” [28]. Based on these facts, several important studies have been conducted, not only to understand these properties attributed to *β*-lapachone, but also to develop them even further, leading to different powerful derivatives with good antitumoral bioactivity [29,30]. For instance, in one specific study conducted by our research group, 3′-nitro-3-phenylamino nor-*β*-lapachone was evaluated against HL60 cancer cells and its mechanism of action was elucidated via experiments involving electrochemical analysis, DNA fragmentation, mitochondrial depolarization, and induced apoptosis/necrosis in HL-60 cells [31]. These preliminary studies have already shed light on the redox-dependent mechanism of quinones and the importance of structural modifications aimed at establishing ingenious alterations in the redox balance of these compounds, thus enabling the development of molecules with potent antitumour activity.

Based on our previous experience in the modification of the A- and C-rings of lapachones aiming at obtaining such bioactive molecules, we have recently combined two quinoidal cores (*ortho*-quinone/*para*-quinone) and selenium-containing quinones (quinone/selenium-containing triazole) for the synthesis of molecules with outstanding activity. The synergistic combination of the two redox centres resulted in compounds with remarkable bioactivity (Figure 1B) [32,33]. This combination not only endows these molecules with good antitumor activity, but it also ensures low cytotoxicity against healthy cell lines, allowing the identification of compounds with better selectivity indexes.

These encouraging results based on our previous strategy have inspired our group to further explore this class of molecules, in the continued effort to identify more potent and superior anti-cancer hybrid molecules. In this connection, another strategy well-explored by our research group is based on the A-ring modification of *para*-quinones aiming at slight alterations in their electrochemical aspects, interfering in the redox balance of these compounds and obviously modelling the electrochemical properties [34,35,36]. These aspects are associated intrinsically with the generation of ROS associated with the antitumour activity of quinones. This strategy can deliver compounds that are more efficient in their ability to kill tumour cells with less damage to healthy cells.

In this context, in the present study, we decided to combine *ortho*-quinones with A-ring modified *para*-quinones, exploring the redox behaviour presented by the distinct quinoidal portions for the identification of compounds with activity against the tumour cell lines (Figure 1C). Click chemistry reactions [37,38,39,40,41,42] have been used for the junction of the two quinone portions and the compounds have been evaluated against nine different cancer cell lines, namely HCT-116, PC3, SNB-19, K-562, HL60, B16, A549, KG1 and RAJI, with L929 cells (non-tumoral mouse fibroblast) used as the control.

**Scheme 1 molecules-28-02222-sch001:**
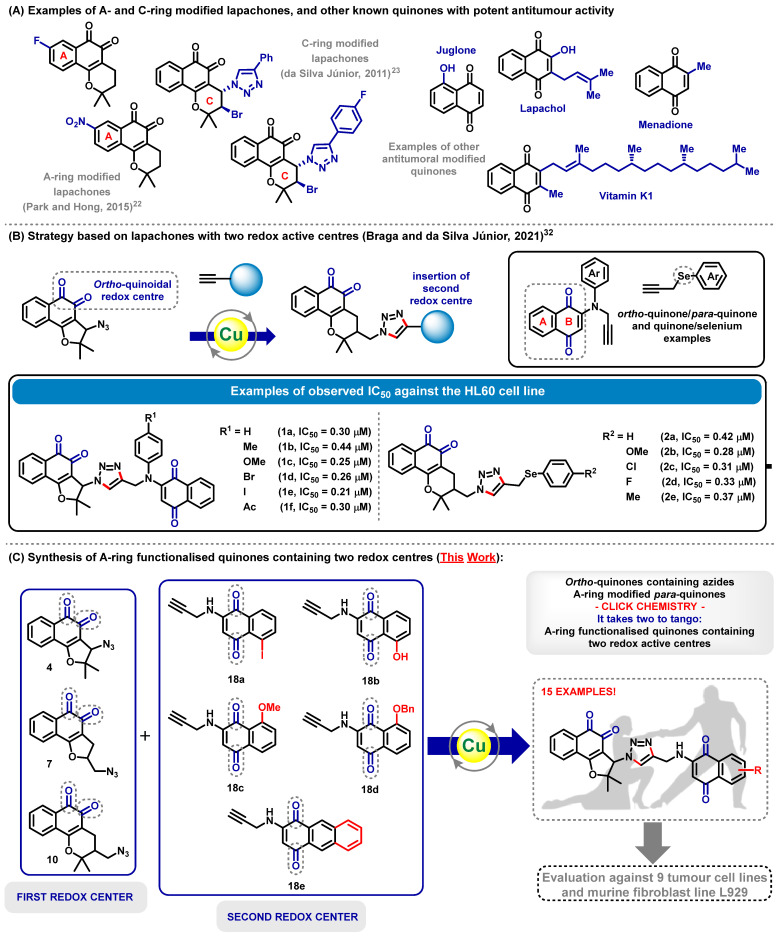
(**A**) Previously reported anticancer A- and C-ring modified β-lapachone derivatives and examples of compounds with antitumour activity [22,23], (**B**) previous work using combined quinones [32] and (**C**) overview of the present work.

## 2. Results and Discussion

### 2.1. Synthesis of the Azide Units

The construction of each family of products was based on the azide-containing quinone side of the molecule. For this matter, three different azides were achieved, all according to their respective sequential synthetic pathway (Figure 2). Azide **4**, the first one to be obtained in this investigation, was originated from *nor*-lapachol (**3**) through a cycloaddition with bromide and a nucleophilic substitution with sodium azide [43]. This process led to the desired quinone (**4**) in the quantitative yield. From this model, another azide (compound **7**) was designed, also according to the previous knowledge of the group [44], starting from a C3-allyl lawsone derivative (**5**). The first step led to the formation of an iodinated 5-membered intermediate (**6)**, in the presence of iodine and pyridine. From this isolated intermediate, a nucleophilic attack with sodium azide results in the desired azide **7**, in a 91% yield. A six-membered azide (compound **10**) may be also achieved when lapachol (**8**) itself is used as a substrate. A sequence of four distinguished steps, passing through an isolatable hydroxylated intermediate (**9**), leads to azide **10** [32], in a 93% yield. In all cases, lapachone products were obtained in good-to-excellent yields.

### 2.2. Synthesis of the Aminoalkyne Units

To react properly with the above-depicted azides, five aminoalkynes (compounds **18a**–**e**) were designed starting from their respective A-ring modified naphthoquinones (**12**–**15**, **17**, Figure 3A). These primordial modifications were also based on previous knowledge of the group, including an aromatic substitution from amine to iodine towards compound **12**, Lewis acid-catalysed nucleophilic substitution aiming compounds **14** and **15** from juglone (**13**), and a reduction/oxidation from quinizarin (**16**) leading to compound **17**.

Once the A-ring modified quinones were achieved, an amination was performed in the presence of propargylamine (Figure 3B), based on a known procedure [45]. This process led to five A-ring modified alkyne quinones, from which interesting bioactive results may be observed once the desired products are accomplished. Since most of the A-ring modified quinones possess a substituent at the C-5 position (with the exception of quinone **17**), the amination procedure can happen on two different sites of the molecule, namely the C-2 or C-3 position, and this difference may generate a mixture of regioisomers. However, although this selectivity was expected to happen, the amination steps led to specific isomers in each case, from which the corresponding regioisomer was observed only as traces, and therefore was not isolated.

To further understand this selectivity, it is important to understand the mechanism by which this reaction happens. The entrance of the aminoalkyne takes place through a nucleophilic attack of the nitrogen atom at one of the two carbons located each at the positions C-2 or C-3 of the B-ring. The selectivity of this attack depends on the relative intensity of the positive charge on each one of these carbon atoms and the corresponding negative charge on the opposite oxygen atom in their resonance contributors. In the case of quinone **12**, it is possible to understand that the polarizability mediated by the large electronic cloud around the iodine made it more reasonable to stabilise a negative charge on vicinal-negative oxygen, consequently increasing the positive charge over the carbon C-2, which becomes more susceptible to a nucleophilic attack. Similar behaviour is expected to happen when the juglone (**13**) itself is used. A negative charge on an oxygen atom near the hydroxyl group can be stabilised easily through a hydrogen bond.

In the case of compounds **14** and **15**, the resonance effect is no longer the main attributor to the observed phenomenon, but the indirect inductive effect instead, over the carbon atoms C-2 and C-3. Through this aspect, the C-3 carbons receive a slightly higher positive partial charge, leading to the C-3 aminated products **18c** and **18d**.

### 2.3. Scopes Achieved

A combination of the previously mentioned azides **4**, **7** and **10** and the aminoalkynes **18a**–**e**, through a copper-catalysed 1,3-dipolar cycloaddition, led to fifteen new triazoles, divided into three families according to the azide applied. This methodology was previously developed [39], and it requires pentahydrated copper sulfate (2 mol%) as a catalyst, and sodium L-ascorbate (5 mol%) as a reducing agent. A mixture of dichloromethane/water (1:1) was found to be a plausible solvent, used here to maximize the solubility of not only the quinoidal substrates, but also the ionic reactants. Therefore, it is important to maintain vigorous stirring during the reaction to provide the surface interaction required between the two phases. The reaction is performed at room temperature, for 24 h. In all cases, it was possible to successfully obtain bi-quinoidal structures presenting two redox centres in moderate-to-good yields.

It is not possible to directly link the structural substituents with their respective reactivity, since the reaction takes place on a site that is not chemically related to the influence of these substituents. It is reasonable to assume that, since it is an interface reaction, the solubility of the compounds involved plays a more important role here.

The first family (**19a**–**e**) was depicted using the azide **4**, originated from *nor*-lapachol (**3**), and the aminoalkyne-quinones **18a**–**e** (Figure 4). In this first family, the best result was obtained when the aminoalkyne **18b** was used, leading to product **19b**, in a 71% yield. This result may be a consequence of the plausible solubility of the reactants (including the substrate **18b**) in both solvents. The opposite behaviour was also observed, when the aminoalkyne **18e** was used. In this particular case, this aminoalkyne does not present a perfect solubility in water or dichloromethane, therefore the product **20e** was achieved in a lower yield (42%) when compared to the rest of the family, but still plausible for the obtention of the quinoidal product presenting two redox centres.

Better results were observed for the construction of the second triazole family (Figure 5). For this case, the azide **7** was combined with the aminoalkynes **18a**–**e**, from which the products **20a**–**e** were achieved in good yields (62–86%).

The third scope involved using the azide **10** combined with the aminoalkynes **18a**–**e** (Figure 6). In this study, the best result was obtained when the aminoalkyne **18a** was used, leading to compound **21a** in a 64% yield. The anthraquinone-derived aminoalkyne **18e**, which previously led to the final product in lower yields, was not different in this case, in which the desired product **21e** was obtained in a 59% yield.

From a general point of view, every substrate submitted to this method led to the desired triazole with success, either in good or lower yields. This fact corroborates the large applicability of this method.

### 2.4. Anticancer Evaluation

Once synthesized and characterized properly, both substrates (azides **4**, **7** and **10**; as well as the aminoalkynes **18a**–**e**) and the triazoles (compounds **19a**–**e**, **20a**–**e** and **21a**–**e**) were evaluated for antitumour activity against nine different cancer cell lines, namely HCT-116 (human colorectal carcinoma), PC3 (human prostate adenocarcinoma), SNB-19 (human astrocytoma), K-562 (human chronic myeloid leukaemia), HL60 (human pro-myelocytic leukaemia), B16 (murine melanoma), A549 (human lung carcinoma), KG1 (human acute myeloid leukaemia) and RAJI (human Burkitt’s lymphoma), with L929 cells (non-tumoral mouse fibroblast) serving as the control (Table 1). L929 cells are required as a benchmark (of toxicity) and for comparison (of selectivity) against the tested cancer cell lines.

In this study, the IC_50_ was obtained in micromolar concentrations, using the colourimetric MTT (3-(4,5-dimethylthiazol-2-yl)-2,5-diphenyl tetrazolium bromide)) assay, and doxorubicin was used as the positive control. The bioactivity was classified according to the IC_50_ value as follows: highly active (IC_50_ < 2 μM), moderately active (2 μM < IC_50_ < 10 μM) and inactive (IC_50_ > 10 μM). In most of the cases, a high-to-moderate activity was observed, especially against the HL60 cell line, for which IC_50_ values as low as 0.3 μM could be successfully achieved (compound **19d**). The activity against the non-tumoural murine fibroblast cell line L929 was also evaluated in order to study the cytotoxicity behaviour of each compound and to understand their respective relative selectivity. The selectivity index was obtained using the ratio of measured cytotoxicity between the L929 cell line and each of the cancer cell lines, and the results are presented in Table 2.

#### 2.4.1. Azide Substrates (**4**, **7** and **10**)

From a general point of view, azides **4** and **7** presented the best activity against all the cell lines studied when compared to azide **10**. Most of the results observed for compounds **5** and **7** were around four times better than the results for azide **10**. However, this result did not negatively affect the activity of the final triazole obtained from azide **10**, since the third family (compounds **21a**–**e**) still presented good results, as can be seen in Section 2.4.5. A very interesting result can be highlighted here, since azide **7**, although moderately active against the PC3 cell line, presented a similar activity against the SNB-19 cell line when compared to the positive control, doxorubicin. Regarding the selectivity, compound **7** presented selectivity indexes near 2.0 related to its activity against the HCT-116, SNB-19, K-562 and B16 cell lines, which basically means that this compound hits these cancer cells twice as hard as non-tumoral cells. When compared to the positive control, doxorubicin (which presents a selectivity index of 1.4 against SNB-19 cells), compound **7** presents an even better selectivity (with an index of 2.4). The similarity of the structures of these azides makes it difficult to propose a direct correlation between structure and reactivity. Furthermore, the results presented by compounds **4** and **7** were similar for most of the cancer cell lines. However, since azide **10** presented a lower activity, it can be inferred that the presence of a six-membered ring might be an issue or an inhibiting factor.

#### 2.4.2. Naphthoquinoidal Aminoalkyne Substrates (**18a**–**e**)

The quinoidal substrates **18a**–**e** did not present potent anticancer activities. In most of the cases, the IC_50_ values obtained were higher than 100 μM. These results, although not satisfactory, are a good example of the synergetic behaviour that quinoidal molecules can present. In most instances, the combination of the unactive naphthoquinoidal aminoalkyne with the previously mentioned azide quinones led to triazole products with a higher activity compared to their respective aminoalkyne precursor. Furthermore, regarding the bioactivity of the aminoalkynes, anticancer activity was observed when compound **18b** was tested against the HCT-116 cancer cell line, presenting an IC_50_ value of 12.73 μM.

#### 2.4.3. First Family of Triazoles (**19a**–**e**)

As a general observation, the first family of triazoles presented the best anticancer activity. Within these results, the best anticancer activities were observed against the HL60 cancer cell line, with IC_50_ values between 0.3 and 1.1 μM. These are impressive results when compared to the positive control, doxorubicin, which, under the same conditions, presented an IC_50_ value of 0.02 μM. Regarding its selectivity, compound **19c** presented good indexes against HCT-16, HL60 and RAJI cell lines (5.8, 8.6 and 9.9 respectively), whereas compound **19b** presented a valuable index of 6.0 against the HL60 cell line. Furthermore, compound **19c** also presented an IC_50_ value of 0.9 μM against the RAJI cell line, more active than the positive control, resulting in the above-mentioned selectivity index of 9.9. Beyond that, combining both activity and selectivity, compound **19d** also presented one of the best performances, with an impressive IC_50_ of 0.3 μM and a selectivity index of 6.7 against the HL60 cell line. With these results in hand and further developments, compounds **19c** and **19d** might indeed become plausible alternatives for the treatment of human Burkitt’s lymphoma and human pro-myelocytic leukaemia, respectively.

#### 2.4.4. Second Family of Triazoles (**20a**–**e**)

The second family of triazoles presented a lower activity when compared to the other two families. Although the results were less impressive in this particular case, compound **20e** can still be highlighted as a prominent molecule, regarding its IC_50_ of 1.6 μM and its selectivity index of 10.1 against the HL60 cell line, being the most active compound in the second family of triazoles. Beyond that, compound **20a** can also be cited here, since it presented moderate anti-cancer properties against all cancer cell lines studied here, and compound **20b**, which presented an IC_50_ value of 2.49 and an impressive selectivity index of 11.1 against the RAJI cell line.

#### 2.4.5. Third Family of Triazoles (**21a**–**e**)

The third family of triazoles provided another good example of the applicability of quinones against the HL60 cell line, since some of its members presented IC_50_ values as low as 0.5 μM. This result was achieved by compound **21e** against the HL60 cell line, leading also to a high selectivity index of 6.6. The selectivity behaviour of this family was similar to the other ones, and impressive results were observed, for instance, for compound **21d**, with an IC_50_ of 0.89 μM against the HL60 cell line and a selectivity index of 5.9.

## 3. Materials and Methods

### 3.1. General Remarks

The solvents were dried using molecular sieves in inert atmosphere storage. Lawsone, *nor*-lapachol (**3**), lapachol (**8**), juglone (**13**), and quinizarin (**16**) were used as purchased without further purification. 5-Amino-1,4-naphthoquinone (**11**) was synthesized according to a procedure already discussed in the literature [46]. The reaction concentration is expressed in molar (M); this concentration was calculated by the ratio of the amount of the main reactant (the limiting agent) in mmol and the volume of the solvent applied in mL. The presented yields refer to isolated compounds, estimated to be >95% pure as determined by ^1^H-NMR. TLC: Merck, TLC Silica gel 60 F_254_, detection at 254 nm. Infrared spectra were recorded on a Bruker ATR FT-IR Alpha device and IR Prestige-21 Shimadzu using KBr plates. Mass-spectra: EI-MS: Jeol AccuTOF at 70 eV; ESI-MS: Bruker maXis and MicrOTOF. High-resolution mass spectrometry (HRMS): Bruker maXis, Bruker MicrOTOF and Jeol AccuTOF. Melting points: Büchi 540 capillary melting point apparatus; values are uncorrected. The NMR spectra were recorded on Avance III HD 400, Avance III 400, and Avance NEO 600 instruments. If not otherwise specified, chemical shifts (*δ*) are provided in ppm. ^13^C-NMR shifts are classified as: C_q_ (non-hydrogenated carbon), CH, CH_2_, and CH_3_, indicating the nature of the carbon assigned, according to what was observed by DEPT or ATP analysis. All of the structure names were given under IUPAC rules by the CS ChemDraw Ultra program. Single crystals were recrystallized from a mixture of acetonitrile and petroleum ether using a system of vapor diffusion. The crystals were analyzed on a XtaLAB Synergy Rigaku four-circle diffractometer. Using Olex2 [47], the structures were solved with the XT [48] structure solution program using Intrinsic Phasing and refined with the XL [49] refinement package using least squares minimization.

### 3.2. Synthesis of Azide Precursors (***4***, ***7***, and ***10***)

3-azido-2,2-dimethyl-2,3-dihydronaphtho[1,2-*b*]furan-4,5-dione (**4**): In a 100 mL rounded-bottom flask, nor-lapachol (3, 456 mg, 2.0 mmol) and DCM (30 mL) were added. The mixture was cooled down to 0 °C, followed by the careful addition of bromine (1.0 mL, 3.12 g, 19.5 mmol). The reaction was kept under continuous stirring at 0 °C for 5 min. The excess bromine, along with the solvent, was removed under reduced pressure, resulting in an orange solid. This mixture was directly used without further purification in the next step through the addition of DCM (10 mL) and sodium azide (390 mg 6.0 mmol). The reaction was kept under continuous stirring at room temperature for 24 h. The final crude was suspended in 15 mL of distilled water, extracted with ethyl acetate (3 × 15 mL), and dried over Na_2_SO_4_. Column chromatography (*n*-hexane/AcOEt 8:2) on silica gel led to the desired azide 4 (538 mg, 100%) as an orange solid. ^1^H NMR (400 MHz, CDCl_3_) *δ* = 8.15–8.13 (m, 1H), 7.72–7.65 (m, 3H), 4.78 (s, 1H), 1.68 (s, 3H), 1.56 (s, 3H). ^13^C NMR (100 MHz, CDCl_3_) *δ* = 180.6 (C_q_), 175.5 (C_q_), 170.4 (C_q_), 134.8 (CH), 133.1 (CH), 131.4 (C_q_), 129.9 (CH), 127.0 (C_q_), 125.4 (CH), 113.8 (C_q_), 95.8 (C_q_), 67.7 (CH), 27.4 (CH_3_), 22.2 (CH_3_). IR (KBr): *ṽ* = 3417, 2965, 2935, 2110, 1697, 1654, 1618, 1571, 1406, 1266, 1217 cm^−1^. m.p. (°C) = 200–202.

The analytical data are in accordance with those reported in the literature [43].

2-allyl-3-hydroxynaphthalene-1,4-dione (**5**): Sodium hydroxide (1.4 g, 35.0 mmol) was dissolved in ethanol (50 mL) in a 250 mL rounded-bottom flask. Lawsone (5.0 g, 29.0 mmol) was added to the mixture, and the final solution was stirred for 1 h at room temperature. The red precipitate (sodium lawsonate) was filtered off, washed with diethyl ether, and dried in a 70 °C oven. The achieved sodium lawsonate (3.0 g, 15 mmol) and allyl bromide (30.0 mL) was added to a 250 mL rounded-bottom flask, and the mixture was stirred at room temperature for 1 h. Distilled water (70 mL) was added, and the final mixture was stirred for a further 24 h at room temperature. The solution was diluted with an additional 30 mL of water and extracted with ethyl acetate (3 × 15 mL). The organic phase was dried over Na_2_SO_4_ and concentrated under reduced pressure. Column chromatography (*n*-hexane/AcOEt 8:2) on silica gel led to the desired product 5 (1.44 g, 45%) as a yellow solid. ^1^H NMR (400 MHz, CDCl_3_) *δ* = 8.11 (dd, *J* = 7.6, 0.4 Hz, 1H), 8.07 (dd, *J* = 7.6, 0.6 Hz, 1H), 7.75 (td, *J* = 7.5, 1.0 Hz, 1H), 7.67 (td, *J* = 7.4, 1.0 Hz, 1H), 7.41 (s, 1H), 5.95–5.85 (m, 1H), 5.17 (dd, *J* = 17.1, 1.4 Hz, 1H), 5.04 (dd, *J* = 10.0, 1.0 Hz, 1H), 3.37 (br s, 1H), 3.35 (br s, 1H). ^13^C NMR (100 MHz, CDCl_3_) *δ* = 184.3 (C_q_), 181.6 (C_q_), 153.3 (C_q_), 135.1 (CH), 133.9 (CH), 133.1 (CH), 132.9 (C_q_), 129.5 (C_q_), 127.0 (CH), 126.3 (CH), 122.0 (C_q_), 116.6 (CH_2_), 27.6 (CH_2_). IR (KBr): *ṽ* = 3355, 1644, 1589, 1371, 1351, 1272, 1230, 729 cm^−1^. m.p. (°C) = 112–113. The analytical data are in accordance with those reported in the literature [50].

2-(iodomethyl)-2,3-dihydronaphtho[1,2-*b*]furan-4,5-dione (**6**): Compound 5 (1.0 g, 5.9 mmol) was dissolved in DCM (100 mL) in a 250 mL rounded-bottom flask. A solution of iodine (7.3 g, 20.0 mmol) in DCM (30 mL) and pyridine (4 mL) was added to the mixture, and the final solution was stirred for 1 h at room temperature, followed by the addition of 100 mL of cold water. The organic phase was separated, washed with a Na_2_CO_3_ 10% solution (3 × 50 mL) and water (3 × 50 mL) and dried over Na_2_SO_4_. Column chromatography (*n*-hexane/AcOEt 8:2) on silica gel led to the desired product 6 (823 mg, 41%) as a red solid. ^1^H NMR (400 MHz, CDCl_3_) *δ* = 8.06 (d, *J* = 7.4 Hz, 1H), 7.68–7.63 (m, 2H), 7.60–7.56 (m, 1H), 5.18–5.11 (m, 1H), 3.51 (s, 1H), 3.49 (d, *J* = 0.8 Hz, 1H), 3.30 (dd, *J* = 16.0, 10.0 Hz, 1H), 2.92 (dd, *J* = 16.0, 6.8 Hz, 1H). ^13^C NMR (100 MHz, CDCl_3_) *δ* = 180.9 (C_q_), 175.4 (C_q_), 169.3 (C_q_), 134.8 (CH), 132.3 (CH), 130.7 (C_q_), 129.7 (CH), 127.3 (C_q_), 124.7 (CH), 115.0 (C_q_), 85.4 (CH), 33.2 (CH_2_), 7.4 (CH_2_). IR (ATR): *ṽ* = 3354, 2366, 1686, 1644, 1609, 1582, 1569, 1408, 1348, 1281, 1240, 1222, 1148, 882, 666 cm^−1^. m.p. (°C) = 145–147. The analytical data are in accordance with those reported in the literature [51].

2-(azidomethyl)-2,3-dihydronaphtho[1,2-*b*]furan-4,5-dione (**7**): Compound 6 (610 mg, 1.8 mmol) and sodium azide (216 mg, 3.3 mmol) were dissolved in DMF (10 mL) in a 50 mL rounded-bottom flask. The mixture was stirred for 12 h at room temperature, followed by extraction with DCM (3 × 15 mL). The organic phase was washed with distilled water (15 mL) and dried over Na_2_SO_4_. Column chromatography (*n*-hexane/AcOEt 8:2) on silica gel led to the desired azide 7 (418 mg, 91%) as an orange solid. ^1^H NMR (400 MHz, CDCl_3_) *δ* = 8.06 (d, *J* = 7.6 Hz, 1H), 7.69–7.64 (m, 2H), 7.62–7.58 (m, 1H), 5.34–5.28 (m, 1H), 3.70–3.60 (m, 2H), 3.26 (dd, *J* = 15.6, 10.4 Hz, 1H), 2.93 (dd, *J* = 15.6, 7.2 Hz, 1H). ^13^C NMR (100 MHz, CDCl_3_) *δ* = 180.8 (C_q_), 175.4 (C_q_), 169.4 (C_q_), 134.8 (CH), 132.3 (CH), 130.6 (C_q_), 129.7 (CH), 127.1 (C_q_), 124.7 (CH), 115.2 (C_q_), 85.8 (CH), 54.2 (CH_2_), 29.5 (CH_2_). IR (KBr): *ṽ* = 3369, 2974, 2105, 1690, 1660, 1588, 1408, 1242, 1216 cm^−1^. m.p. (°C) = 172–174. The analytical data are in accordance with those reported in the literature [44].

3-(hydroxymethyl)-2,2-dimethyl-3,4-dihydro-2*H*-benzo[*h*]chromene-5,6-dione (**9**): Formic acid (5.0 mL) was placed in a 50 mL rounded-bottom flask and heated until reaching 90 °C. Paraformaldehyde (264 mg, 8.9 mmol) and lapachol (8, 1.1 g, 4.4 mmol) were added, and the mixture was stirred at 90 °C for 2 h. Distilled water (10 mL) was added to the solution and the reaction was kept under reflux for an additional 12 h. The solution was cooled to room temperature and neutralized with Na_2_CO_3_ (7.4 g) carefully added. The mixture was extracted with ethyl acetate (3 × 150 mL) and the organic phase was dried over Na_2_SO_4_. Column chromatography (*n*-hexane/AcOEt 8:2) on silica gel led to the desired product 9 (755 mg, 63%) as an orange solid. ^1^H NMR (400 MHz, CDCl_3_) *δ* = 8.00 (d, *J* = 7.6 Hz, 1H), 7.78 (d, *J* = 8.0 Hz, 1H), 7.62 (t, *J* = 7.6 Hz, 1H), 7.48 (t, *J* = 7.2 Hz, 1H), 3.86 (dd, *J* = 11.2, 5.2 Hz, 1H), 3.63 (dd, *J* = 10.8, 7.2 Hz, 1H), 2.78 (dd, *J* = 17.6, 5.6 Hz, 1H), 2.51 (br s, 1H), 2.31 (dd, *J* = 17.6, 10.0 Hz, 1H), 2.07–2.00 (m, 1H), 1.60 (s, 3H), 1.34 (s, 3H). ^13^C NMR (400 MHz, CDCl_3_) *δ* = 179.9 (C_q_), 178.6 (C_q_), 162.0 (C_q_), 135.0 (CH), 132.4 (C_q_), 130.9 (CH), 130.2 (C_q_), 128.7 (CH), 124.3 (CH), 112.7 (C_q_), 81.9 (C_q_), 63.0 (CH_2_), 42.6 (CH), 27.8 (CH_3_), 22.2 (CH_3_), 19.8 (CH_2_). IR (KBr): *ṽ* = 3519, 3464, 2981, 2933, 1695, 1648, 1602, 1571, 1398, 1126 cm^−1^. m.p. (°C) = 145–148. The analytical data are in accordance with those reported in the literature [32].

3-(azidomethyl)-2,2-dimethyl-3,4-dihydro-2*H*-benzo[*h*]chromene-5,6-dione (**10**): Compound 9 (272 mg, 1.0 mmol) was dissolved in DCM (10 mL) at 0 °C in a 25 mL rounded-bottom flask. Triethylamine (280 μL, 2.0 mmol) and methanesulfonyl chloride (120 μL, 1.4 mmol) were added to the solution, which was stirred for 30 min at 0 °C. The solvent was removed under reduced pressure and redissolved in DMF (10 mL). Sodium azide (200 mg, 3.1 mmol) was added, and the final mixture was stirred for an additional 48 h at room temperature, followed by extraction with ethyl acetate (3 × 150 mL). The organic phase was dried over Na_2_SO_4_. Column chromatography (*n*-hexane/AcOEt 2:1) on silica gel led to the desired azide 10 (276 mg, 93%) as a dark orange solid. ^1^H NMR (400 MHz, CDCl_3_) *δ* = 8.04 (d, *J* = 7.6 Hz, 1H), 7.77 (d, *J* = 7.6 Hz, 1H), 7.64 (dt, *J* = 7.6, 1.2 Hz, 1H), 7.50 (dt, *J* = 7.6, 0.8 Hz, 1H), 3.56 (dd, *J* = 14.4, 5.2 Hz, 1H), 3.24 (dd, *J* = 12.4, 8.0, 1H), 2.80 (dd, *J* = 18.0, 5.6 Hz, 1H), 2.32 (dd, *J* = 18.0, 9.6 Hz, 1H), 2.08–2.01 (m, 1H), 1.58 (s, 3H), 1.35 (s, 3H). ^13^C NMR (100 MHz, CDCl_3_) *δ* = 179.6 (C_q_), 178.5 (C_q_), 161.5 (C_q_), 135.0 (CH), 132.1 (C_q_), 131.0 (CH), 130.2 (C_q_), 128.8 (CH), 124.2 (CH), 112.0 (C_q_), 80.9 (C_q_), 52.3 (CH_2_), 40.1 (CH), 27.4 (CH_3_), 22.1 (CH_3_), 20.8 (CH_2_). IR (KBr): *ṽ* = 3431, 2928, 2097, 1693, 1606, 1589, 1392, 1261, 1231, 1130 cm^−1^. m.p. (°C) = 103–106. The analytical data are in accordance with those reported in the literature [32].

### 3.3. Synthesis of A-Ring-Modified Quinoidal Substrates

5-Iodo-1,4-naphthoquinone (**12**): 5-Amino-1,4-naphthoquinone (11, 1.0 g, 5.77 mmol) and glacial acetic acid (31.4 mL) were placed in a 250 mL rounded-bottom flask under continuous stirring at room temperature. A mixture of sulfuric acid/water 2:1 (24 mL) was carefully added, and the final mixture was transferred to a 250 mL beaker, with extra care for complete removal of the residual solid. A solution of sodium nitrite (600 mg, 8.65 mmol) in 1.0 mL of water was added to the reaction under continuous stirring at 0 °C. The obtained solution was then quickly converted onto a solution of potassium iodide (3.0 g, 17.9 mmol) in distilled water (80 mL) in a 1.0 L beaker. The reaction was kept under stirring at 90 °C for 20 min. After completion of the reaction, the final mixture was kept at −22 °C in a fridge for 18 h, from which a precipitate was formed and, subsequentially, filtered off. Column chromatography (silica gel, toluene) led to the obtention of 5-iodo-1,4-naphthoquinone (12, 500 mg, 30%) as a red solid. ^1^H NMR (400 MHz, CDCl_3_) *δ* = 8.36 (d, *J *= 7.8 Hz, 1H), 8.15 (d, *J *= 7.8 Hz, 1H), 7.35 (t, *J *= 7.8 Hz, 1H), 7.02 (d, *J *= 10.3 Hz, 1H), 6.94 (d, *J *= 10.3 Hz, 1H). ^13^C NMR (100 MHz, CDCl_3_) *δ* = 183.7 (C_q_), 183.3 (C_q_), 148.3 (CH), 139.9 (CH), 137.2 (CH), 134.4 (C_q_), 133.8 (CH), 130.8 (C_q_), 127.7 (CH), 92.9 (C_q_). IR (ATR): *ṽ *= 1665, 1613, 1567, 1319, 782, 563 cm^−1^. m.p. (°C) = 171–172; HRMS (ESI): Calcd. for C_10_H_6_IO_2_ [M+H]^+^ 284.9407, found 284.9412. The analytical data are in accordance with those reported in the literature [36].

5-Methoxy-1,4-naphthoquinone (**14**): Juglone (13, 174 mg, 1.0 mmol), iodomethane (125 µL, 285 mg, 2.0 mmol) and Ag_2_O (463 mg, 2.0 mmol) were dissolved in dichloromethane (30 mL) in a 125 mL rounded-bottom flask. The solution was kept under reflux for 48 h. The final solution was filtered through a pad of celite and washed with dichloromethane. The solution was then concentrated under reduced pressure and purified through column chromatography (*n*-hexane/AcOEt 8:2) to provide 5-methoxy-1,4-naphthoquinone (14, 147 mg, 78%) as a yellow solid. ^1^H NMR (400 MHz, CDCl_3_) *δ* = 7.72–7.65 (m, 2H), 7.30 (dd, *J* = 8.1, 1.3 Hz, 1H), 6.87–6.82 (m, 2H), 3.99 (s, 3H). ^13^C NMR (100 MHz, CDCl_3_) *δ* = 185.4 (C_q_), 184.5 (C_q_), 159.8 (C_q_), 141.0 (CH), 136.4 (CH), 135.2 (CH), 134.2 (C_q_), 119.9 (C_q_), 119.3 (CH), 118.1 (CH), 56.7 (CH_3_). IR (ATR): *ṽ *= 1651, 1613, 1581, 1469, 1442, 1376, 1296 cm^−1^. m.p. (°C) = 180–182; HRMS (ESI): Calcd. for C_11_H_9_O_3_ [M+H]^+^ 189.0546, found 189.0546. The analytical data are in accordance with those reported in the literature [52].

5-Benzyloxy-1,4-naphthoquinone (**15**): Juglone (13, 174 mg, 1.0 mmol), benzyl bromide (513 mg, 3.0 mmol), and Ag_2_O (463 mg, 2.0 mmol) were dissolved in dichloromethane (30 mL) in a 125 mL rounded-bottom flask. The solution was kept under stirring at 25 °C for 24 h. The final solution was filtered through a pad of celite and washed with dichloromethane. The solution was then concentrated under reduced pressure and purified through column chromatography (*n*-hexane/AcOEt 8:2) to provide 5-methoxy-1,4-naphthoquinone (15, 137 mg, 52%) as a red solid. ^1^H NMR (400 MHz, CDCl_3_) *δ* = 7.71 (dd, *J* = 7.6, 0.8 Hz, 1H), 7.62 (t, *J* = 8.4 Hz, 1H), 7.57 (d, *J* = 7.2 Hz, 2H), 7.40 (t, *J* = 7.2 Hz, 2H), 7.33–7.31 (m, 2H), 6.86, (s, 2H), 5.27 (s, 2H). ^13^C NMR (100 MHz, CDCl_3_) *δ* = 185.3 (C_q_), 184.2 (C_q_), 158.6 (C_q_), 140.9 (CH), 136.3 (CH), 136.1 (C_q_), 134.9 (CH), 134.2 (C_q_), 128.8 (CH), 128.0 (CH), 126.7 (CH), 120.3 (C_q_), 119.7 (CH), 119.5 (CH), 70.9 (CH_2_). IR (KBr): *ṽ *= 1747, 1660, 1614, 1582, 1497, 1454, 1254, 1023, 733, 697 cm^−1^. m.p. (°C) = 30–32; HRMS (ESI): Calcd. for C_17_H_12_O_3_Na [M+Na]^+^ 287.0679, found 287.0677. The analytical data are in accordance with those reported in the literature [53,54].

1,4-Antraquinone (**17**): Quinizarin (16, 989 mg, 4.12 mmol) was dissolved in methanol (19 mL) at 0 °C. Sodium borohydride (945 mg, 25.0 mmol) was added carefully. The reaction was stirred for 90 min at 0 °C. An aq. solution of hydrochloric acid (6 m, 18 mL) was added, and the precipitate was filtered off and washed with water to afford 1,4-anthraquinone 17 as a brown solid (791 mg, 92%). ^1^H NMR (400 MHz, CDCl_3_) *δ* = 8.55 (s, 2H), 8.01 (s, 2H), 7.66 (s, 2H), 7.03 (s, 2H). ^13^C NMR (100 MHz, CDCl_3_) *δ* = 184.8 (C_q_), 140.2 (CH), 134.9 (C_q_), 130.4 (CH), 129.8 (CH), 129.0 (CH), 128.5 (C_q_). IR (ATR): *ṽ *= 3052, 1665, 1614, 1596, 1448, 1293 cm^−1^. m.p. (°C) = 212–216. HRMS (ESI): Calcd. for C_14_H_9_O_2_ [M+H]^+^ 209.0597, found 209.0604. The analytical data are in accordance with those reported in the literature [55].

### 3.4. General Procedure for the Synthesis of Amino-Alkynes (***18a**–**e***)

The corresponding quinone (1.0 mmol) was dissolved in acetonitrile (3.0 mL, 0.3 m) at room temperature in a 10 mL rounded-bottom flask. *N*-propargylamine (128 μL, 110.2 mg, 2.0 mmol) was added to the mixture and it was kept under continuous stirring over 24 h at room temperature. The respective amino-alkyne was obtained by column chromatography (*n*-hexane/EtOAc 8:2). The correct position of the propargylamine substituent was determined over bidimensional NMR spectra analysis.

5-iodo-2-(prop-2-yn-1-ylamino)naphthalene-1,4-dione (**18a**): The general procedure for the synthesis of amino-alkynes was followed using 5-iodo-1,4-naphthoquinone (12, 284 mg, 1.0 mmol) and *N*-propargylamine (128 μL, 110.2 mg, 2.0 mmol). Purification by column chromatography on silica gel (*n*-hexane/EtOAc 8:2) yielded 8-iodo-2-(prop-2-yn-1-ylamino)naphthalene-1,4-dione (18a, 243 mg, 72%) as an orange solid. ^1^H NMR (400 MHz, DMSO-*d*_6_) *δ* = 8.31 (dd, *J* = 8.0, 1.2 Hz, 1H), 8.05 (dd, *J* = 7.6, 0.8 Hz, 1H), 7.97 (t, *J* = 6.0 Hz, 1H), 7.47 (t, *J* = 7.6 Hz, 1H), 5.79 (s, 1H), 4.05 (dd, *J* = 6.0, 2.4 Hz, 2H), 3.27 (t, *J* = 2.4 Hz, 1H). ^13^C NMR (100 MHz, DMSO-*d*_6_) *δ* = 180.2 (C_q_), 180.1 (C_q_), 148.9 (C_q_), 146.6 (CH), 136.0 (C_q_), 135.2 (CH), 130.0 (C_q_), 126.8 (CH), 100.9 (CH), 94.6 (C_q_), 79.3 (C_q_), 75.1 (CH), 31.7 (CH_2_). IR (KBr): *ṽ *= 3371, 3280, 1673, 1600, 1494, 1250, 660 cm^−1^. m.p. (°C) = 44–49.

5-hydroxy-2-(prop-2-yn-1-ylamino)naphthalene-1,4-dione (**18b**): The general procedure for the synthesis of amino-alkynes was followed using Juglone (13, 174 mg, 1.0 mmol) and *N*-propargylamine (128 μL, 110.2 mg, 2.0 mmol). Purification by column chromatography on silica gel (*n*-hexane/EtOAc 8:2) yielded 8-hydroxy-2-(prop-2-yn-1-ylamino)naphthalene-1,4-dione (**18b**, 80 mg, 35%) as a red solid. ^1^H NMR (400 MHz, DMSO-*d*_6_) *δ *= 13.19 (s, 1H), 8.20 (br s, 1H), 7.61 (t, *J* = 7.6 Hz, 1H), 7.52 (d, *J* = 7.6 Hz, 1H), 7.29 (d, *J* = 8.4 Hz, 1H), 5.74 (s, 1H), 4.08 (d, *J* = 3.6 Hz, 2H), 3.29 (s, 1H). ^13^C NMR (100 MHz, DMSO-*d*_6_) *δ *= 188.2 (C_q_), 180.5 (C_q_), 160.1 (C_q_), 148.9 (C_q_), 134.4 (CH), 130.4 (C_q_), 125.1 (CH), 118.5 (CH), 114.1 (C_q_), 100.0 (CH), 78.4 (C_q_), 74.8 (CH), 31.2 (CH_2_). IR (KBr): *ṽ *= 3348, 3296, 2917, 2358, 2340, 1600, 1616, 1471, 1249, 1225 cm^−1^. m.p. (°C) = 45–51. The analytical data are in accordance with those reported in the literature [45]. The structure of the product was also confirmed by X-ray diffraction (CCDC number = 2226471).

8-methoxy-2-(prop-2-yn-1-ylamino)naphthalene-1,4-dione (**18c**): The general procedure for the synthesis of amino-alkynes was followed using 5-methoxy-1,4-naphthoquinone (14, 188 mg, 1.0 mmol) and *N*-propargylamine (128 μL, 110.2 mg, 2.0 mmol). Purification by column chromatography on silica gel (*n*-hexane/EtOAc 8:2) yielded 8-methoxy-2-(prop-2-yn-1-ylamino)naphthalene-1,4-dione (**18c**, 125 mg, 52%) as a red solid. ^1^H NMR (400 MHz, DMSO-*d*_6_) *δ *= 7.77 (t, *J* = 8.0 Hz, 1H), 7.66 (t, *J* = 6.0 Hz, 1H), 7.59 (dd, *J* = 7.2, 0.4 Hz, 1H), 7.42 (d, *J* = 8.0 Hz, 1H), 5.70 (s, 1H), 4.04 (dd, *J* = 6.0, 2.0 Hz, 2H), 3.93 (s, 3H), 3.26 (t, *J* = 2.3 Hz, 1H). ^13^C NMR (100 MHz, DMSO-*d*_6_) *δ *= 181.2 (C_q_), 179.3 (C_q_), 159.7 (C_q_), 148.9 (C_q_), 136.1 (CH), 135.1 (C_q_), 117.9 (CH), 117.8 (C_q_), 117.0 (CH), 99.7 (CH), 79.1 (C_q_), 74.5 (CH), 56.4 (CH_3_), 31.2 (CH_2_). IR (KBr): *ṽ *= 3339, 3203, 2941, 1674, 1608, 1577, 1261, 1217 cm^−1^. m.p. (°C) = 52–57. The analytical data are in accordance with those reported in the literature [45]. The structure of the product was also confirmed by X-ray diffraction (CCDC number = 2226469).

8-benzyloxy-2-(prop-2-yn-1-ylamino)naphthalene-1,4-dione (**18d**): The general procedure for the synthesis of amino-alkynes was followed using 5-benzyloxy-1,4-naphthoquinone (15, 264 mg, 1.0 mmol) and *N*-propargylamine (128 μL, 110.2 mg, 2.0 mmol). Purification by column chromatography on silica gel (*n*-hexane/EtOAc 8:2) yielded 8-benzyloxy-2-(prop-2-yn-1-ylamino)naphthalene-1,4-dione (**18d**, 155 mg, 49%) as a brown solid. ^1^H NMR (400 MHz, DMSO-*d*_6_) *δ *= 7.82 (t, *J* = 6.0 Hz, 1H), 7.76 (t, *J* = 8.0 Hz, 1H), 7.61–7.59 (m, 3H), 7.50 (d, *J* = 8.4 Hz, 1H), 7.41 (t, *J* = 7.2 Hz, 2H), 7.33 (t, *J* = 7.6 Hz, 1H), 5.69 (s, 1H), 5.30 (s, 2H), 4.01 (dd, *J* = 5.6, 2.0 Hz, 2H), 3.23 (t, *J* = 2.0 Hz, 1H). ^13^C NMR (100 MHz, DMSO-*d*_6_) *δ *= 181.3 (C_q_), 179.5 (C_q_), 158.6 (C_q_), 149.0 (C_q_), 136.7 (CH), 136.1 (C_q_), 135.2 (C_q_), 128.4 (CH), 127.7 (CH), 127.0 (CH), 118.2 (CH), 118.2 (CH), 118.2 (C_q_), 99.8 (CH), 79.1 (C_q_), 74.5 (CH), 70.1 (CH_2_), 31.2 (CH_2_). IR (KBr): *ṽ *= 3359, 3289, 2359, 1674, 1601, 1577, 1494, 1252, cm^−1^. m.p. (°C) = 50–55. The analytical data are in accordance with those reported in the literature [45]. The structure of the product was also confirmed by X-ray diffraction (CCDC number = 2226470).

2-(prop-2-yn-1-ylamino)anthracene-1,4-dione (**18e**): The general procedure for the synthesis of amino-alkynes was followed using 1,4-antraquinone (17, 208 mg, 1.0 mmol) and *N*-propargylamine (128 μL, 110.2 mg, 2.0 mmol). Purification by column chromatography on silica gel (*n*-hexane/EtOAc 8:2) yielded 2-(prop-2-yn-1-ylamino)anthracene-1,4-dione (**18e**, 167 mg, 64%) as a brown solid. ^1^H NMR (400 MHz, DMSO-*d*_6_) *δ *= 8.62 (s, 1H), 8.50 (s, 1H), 8.21 (dd, *J* = 13.6, 7.6 Hz, 2H), 7.88 (t, *J* = 5.6 Hz, 1H), 7.75–7.68 (m, 2H), 5.89 (s, 1H), 4.06 (d, *J* = 4.0 Hz, 2H), 3.25 (br s, 1H). ^13^C NMR (100 MHz, DMSO-*d*_6_) *δ *= 181.5 (C_q_), 181.0 (C_q_), 149.0 (C_q_), 135.1 (C_q_), 133.7 (C_q_), 130.3 (CH), 130.0 (CH), 129.9 (CH), 129.3 (C_q_), 129.1 (CH), 128.7 (CH), 127.5 (C_q_), 126.9 (CH), 103.4 (CH), 79.1 (C_q_), 74.7 (CH), 31.3 (CH_2_). IR (KBr): *ṽ *= 3356, 3217, 2922, 2359, 1668, 1598, 1509, 1320, 1263 cm^−1^. m.p. (°C) = 42–47.

### 3.5. General Procedure for Triazole Synthesis via a Copper-Catalysed 1,3-Dipolar Cycloaddition (***19a**–**21e***)

A reaction tube was charged with the corresponding azide-lapachone derivative (0.2 mmol, 1.0 equiv.), amino-alkyne naphthoquinone (0.22 mmol, 1.1 equiv.), CuSO_4_⋅5H_2_O (1 mg, 0.002 mmol, 1 mol %), and sodium ascorbate (4 mg, 0.01 mmol, 5 mol %). Then, a mixture DCM/distilled water 1:1 (4 mL) was added. The reaction mixture was kept under vigorous stirring at room temperature for 24 h. The crude product was partitioned with distilled water (20 mL) and DCM (3 × 30 mL). The organic phase was dried over Na_2_SO_4_, filtered and purified by column chromatography (*n*-hexane/EtOAc 7:3).

### 3.6. Characterization Data of Products ***19a**–**21e***

3-(4-(((5-iodo-1,4-dioxo-1,4-dihydronaphthalen-2-yl)amino)methyl)-1*H*-1,2,3-triazol-1-yl)-2,2-dimethyl-2,3-dihydronaphtho[1,2-*b*]furan-4,5-dione (**19a**): The general procedure for triazole synthesis was followed by using 3-azido-2,2-dimethyl-2,3-dihydronaphtho[1,2-*b*]furan-4,5-dione (**4**) (54 mg, 0.2 mmol) and 5-iodo-2-(prop-2-yn-1-ylamino)naphthalene-1,4-dione (**18a**) (74 mg, 0.22 mmol) as the starting materials. Purification by column chromatography on silica gel (*n*-hexane/EtOAc 4:1) yielded **19a** (74 mg, 61%) as an orange solid. ^1^H NMR (600 MHz, DMSO-*d_6_*) *δ *= 8.29 (s, 1H), 8.04 (d, *J* = 7.2 Hz, 1H), 8.00–7.90 (m, 2H), 7.84–7.78 (m, 3H), 7.75 (d, *J* = 7.2 Hz, 1H), 7.72 (td, 7.2, 0.6 Hz, 1H), 6.04 (s, 1H), 5.70 (d, *J* = 6.0 Hz, 1H), 4.48 (d, *J* = 6.0 Hz, 2H), 1.68 (s, 3H), 1.00 (s, 3H). ^13^C NMR (150 MHz, DMSO-*d_6_*) *δ *= 181.7 (C_q_), 180.0 (C_q_), 179.6 (C_q_), 174.7 (C_q_), 170.0 (C_q_), 148.9 (C_q_), 148.4 (C_q_), 146.3 (CH), 143.2 (C_q_), 135.1 (C_q_), 134.9 (CH), 133.2 (CH), 128.9 (CH), 125.5 (CH), 125.2 (CH), 123.6 (CH), 111.3 (C_q_), 100.7 (CH), 99.8 (C_q_), 95.5 (C_q_), 94.3 (C_q_), 66.2 (CH), 37.8 (CH_2_), 37.7 (C_q_), 27.1 (CH_3_), 22.7 (CH), 20.7 (CH_3_). IR (KBr): *ṽ *= 3365, 2360, 1657, 1609, 1570, 1508, 1221, 1082, 662 cm^−1^. m.p. (°C) = 112–116. HRMS (ESI): Calcd. for C_27_H_20_IN_4_O_5_ [M+H]^+^ 607.0478, found 607.0459.

3-(4-(((5-hydroxy-1,4-dioxo-1,4-dihydronaphthalen-2-yl)amino)methyl)-1*H*-1,2,3-triazol-1-yl)-2,2-dimethyl-2,3-dihydronaphtho[1,2-*b*]furan-4,5-dione (**19b**): The general procedure for triazole synthesis was followed by using 3-azido-2,2-dimethyl-2,3-dihydronaphtho[1,2-*b*]furan-4,5-dione (**4**) (54 mg, 0.2 mmol) and 5-hydroxy-2-(prop-2-yn-1-ylamino)naphthalene-1,4-dione (**18b**) (50 mg, 0.22 mmol) as the starting materials. Purification by column chromatography on silica gel (*n*-hexane/EtOAc 4:1) yielded **19b** (71 mg, 71%) as an orange solid. ^1^H NMR (600 MHz, DMSO-*d_6_*) *δ *= 13.24 (s, 1H), 8.34 (t, *J* = 6.0 Hz, 1H), 8.28 (s, 1H), 8.05 (d, *J* = 7.2 Hz, 1H), 7.84 (t, *J* = 7.8 Hz, 1H), 7.80 (t, *J* = 7.8 Hz, 1H), 7.76 (d, *J* = 7.2 Hz, 1H), 7.58 (t, *J* = 7.8 Hz, 1H), 7.50 (d, *J* = 7.8 Hz, 1H), 7.27 (d, *J* = 8.4 Hz, 1H), 6.03 (s, 1H), 5.68 (s, 1H), 4.51 (d, *J* = 6.0 Hz, 2H), 1.68 (s, 3H), 1.00 (s, 3H). ^13^C NMR (150 MHz, DMSO-*d_6_*) *δ *= 188.2 (C_q_), 180.8 (C_q_), 179.9 (C_q_), 174.6 (C_q_), 169.9 (C_q_), 160.3 (C_q_), 149.5 (C_q_), 142.8 (C_q_), 134.9 (CH), 134.5 (CH), 133.2 (CH), 131.8 (C_q_), 130.6 (C_q_), 128.9 (CH), 126.6 (C_q_), 125.3 (CH), 125.1 (CH), 123.7 (CH), 118.7 (CH), 114.4 (C_q_), 111.3 (C_q_), 99.4 (CH), 95.4 (C_q_), 66.2 (CH), 40.1 (C_q_), 37.8 (CH_2_), 27.0 (CH_3_), 20.7 (CH_3_). IR (KBr): *ṽ *= 3348, 2358, 2340, 1652, 1616, 1569, 1469, 1248, 1051 cm^−1^. m.p. (°C) = 114–118. HRMS (ESI): Calcd. for C_27_H_20_N_4_O_6_Na [M+Na]^+^ 519.1281, found 519.1282.

3-(4-(((8-methoxy-1,4-dioxo-1,4-dihydronaphthalen-2-yl)amino)methyl)-1*H*-1,2,3-triazol-1-yl)-2,2-dimethyl-2,3-dihydronaphtho[1,2-*b*]furan-4,5-dione (**19c**): The general procedure for triazole synthesis was followed by using 3-azido-2,2-dimethyl-2,3-dihydronaphtho[1,2-*b*]furan-4,5-dione (**4**) (54 mg, 0.2 mmol) and 8-methoxy-2-(prop-2-yn-1-ylamino)naphthalene-1,4-dione (**18c**) (53 mg, 0.22 mmol) as the starting materials. Purification by column chromatography on silica gel (*n*-hexane/EtOAc 4:1) yielded **19c** (59 mg, 58%) as an orange solid. ^1^H NMR (600 MHz, DMSO-*d_6_*) *δ *= 8.27 (s, 1H), 8.03 (d, *J* = 7.2 Hz, 1H), 7.83 (t, *J* = 7.2 Hz, 1H), 7.78 (t, *J* = 7.8 Hz, 1H), 7.75–7.72 (m, 3H), 7.54 (d, *J* = 7.2 Hz, 1H), 7.38 (d, *J* = 8.4 Hz, 1H), 6.03 (s, 1H), 5.59 (s, 1H), 4.45 (d, *J* = 6.0 Hz, 2H), 3.88 (s, 3H), 1.67 (s, 3H), 0.99 (s, 3H). ^13^C NMR (150 MHz, DMSO-*d_6_*) *δ *= 181.1 (C_q_), 179.9 (C_q_), 179.4 (C_q_), 174.6 (C_q_), 169.9 (C_q_), 159.7 (C_q_), 149.2 (C_q_), 136.2 (CH), 135.2 (C_q_), 134.8 (CH), 133.1 (CH), 131.8 (C_q_), 128.8 (CH), 126.6 (C_q_), 125.1 (CH), 123.5 (CH), 117.9 (CH), 117.8 (C_q_), 117.0 (CH), 111.3 (C_q_), 99.0 (CH), 95.4 (C_q_), 66.2 (CH), 56.4 (CH_3_), 40.1 (C_q_), 37.6 (CH_2_), 27.0 (CH_3_), 20.7 (CH_3_). IR (KBr): *ṽ *= 3437, 3238, 2360, 2341, 1667, 1653, 1615, 1572, 1384, 1220, 1048 cm^−1^. m.p. (°C) = 173–179. HRMS (ESI): Calcd. for C_28_H_22_N_4_O_6_Na [M+Na]^+^ 533.1437, found 533.1425.

3-(4-(((8-benzyloxy-1,4-dioxo-1,4-dihydronaphthalen-2-yl)amino)methyl)-1*H*-1,2,3-triazol-1-yl)-2,2-dimethyl-2,3-dihydronaphtho[1,2-*b*]furan-4,5-dione (**19d**): The general procedure for triazole synthesis was followed by using 3-azido-2,2-dimethyl-2,3-dihydronaphtho[1,2-*b*]furan-4,5-dione (**4**) (54 mg, 0.2 mmol) and 8-benzyloxy-2-(prop-2-yn-1-ylamino)naphthalene-1,4-dione (**18d**) (70 mg, 0.22 mmol) as the starting materials. Purification by column chromatography on silica gel (*n*-hexane/EtOAc 4:1) yielded **19d** (57 mg, 49%) as an orange solid. ^1^H NMR (600 MHz, DMSO-*d_6_*) *δ *= 8.25 (s, 1H), 8.04 (d, *J* = 7.2 Hz, 1H), 7.92 (t, *J* = 6.0 Hz, 1H), 7.85–7.74 (m, 4H), 7.60 (d, *J* = 7.2 Hz, 2H), 7.57 (d, *J* = 7.8 Hz, 1H), 7.50 (d, *J* = 8.4 Hz, 1H), 7.41 (t, *J* = 7.2 Hz, 2H), 7.33 (t, *J* = 7.2 Hz, 1H), 6.02 (s, 1H), 5.62 (s, 1H), 5.30 (s, 2H), 4.44 (d, *J* = 5.4 Hz, 2H), 1.67 (s, 3H), 0.98 (s, 3H). ^13^C NMR (150 MHz, DMSO-*d_6_*) *δ *= 181.0 (C_q_), 179.8 (C_q_), 179.6 (C_q_), 174.6 (C_q_), 169.9 (C_q_), 158.6 (C_q_), 149.2 (C_q_), 143.2 (C_q_), 136.7 (C_q_), 136.1 (CH), 135.3 (C_q_), 134.8 (CH), 133.1 (CH), 131.8 (C_q_), 128.8 (CH), 128.4 (CH), 127.7 (CH), 127.0 (CH), 126.6 (C_q_), 125.1 (CH), 123.5 (CH), 118.2 (CH), 118.1 (CH), 111.3 (C_q_), 99.0 (CH), 95.3 (C_q_), 70.1 (CH_2_), 66.1 (CH), 40.1 (C_q_), 37.6 (CH_2_), 27.0 (CH_3_), 20.6 (CH_3_). IR (KBr): *ṽ *= 3372, 2362, 1651, 1609, 1571, 1497, 1356, 1279, 1053 cm^−1^. m.p. (°C) = 205–210. HRMS (ESI): Calcd. for C_34_H_27_N_4_O_6_ [M+H]^+^ 587.1931, found 587.1929.

3-(4-(((1,4-dioxo-1,4-dihydroanthracen-2-yl)amino)methyl)-1*H*-1,2,3-triazol-1-yl)-2,2-dimethyl-2,3-dihydronaphtho[1,2-*b*]furan-4,5-dione (**19e**): The general procedure for triazole synthesis was followed by using 3-azido-2,2-dimethyl-2,3-dihydronaphtho[1,2-*b*]furan-4,5-dione (**4**) (54 mg, 0.2 mmol) and 2-(prop-2-yn-1-ylamino)anthracene-1,4-dione (**18e**) (58 mg, 0.22 mmol) as the starting materials. Purification by column chromatography on silica gel (*n*-hexane/EtOAc 4:1) yielded **19e** (45 mg, 42%) as an orange solid. ^1^H NMR (600 MHz, DMSO-*d_6_*) *δ *= 8.60 (br s, 1H), 8.45 (br s, 1H), 8.31 (br s, 1H), 8.21 (d, *J* = 7.6 Hz, 1H), 8.17 (d, *J* = 7.6 Hz, 1H), 8.02 (d, *J* = 7.1 Hz, 1H), 7.99 (t, *J* = 4.9 Hz, 1H), 7.81–7.76 (m, 2H), 7.73–7.68 (m, 3H), 6.03 (s, 1H), 5.82 (s, 1H), 4.50 (d, *J* = 5.3 Hz, 2H), 1.67 (s, 3H), 1.00 (s, 3H). ^13^C NMR (100 MHz, DMSO-*d_6_*) *δ *= 181.5 (C_q_), 180.2 (C_q_), 175.0 (C_q_), 170.2 (C_q_), 149.6 (C_q_), 143.5 (C_q_), 135.5 (C_q_), 135.2 (CH), 135.2 (C_q_), 134.0 (C_q_), 133.5 (CH), 132.2 (C_q_), 130.6 (CH), 130.2 (CH), 129.8 (C_q_), 129.3 (CH), 129.2 (CH), 128.9 (CH), 127.9 (C_q_), 127.1 (CH), 127.0 (CH), 125.5 (CH), 124.0 (CH), 111.7 (C_q_), 103.0 (CH), 95.7 (C_q_), 66.6 (CH), 60.3 (C_q_), 38.1 (CH_2_), 27.4 (CH_3_), 21.1 (CH_3_). IR (KBr): *ṽ *= 3374, 2923, 2852, 2358, 1658, 1601, 1572, 1313, 1261, 1084 cm^−1^. m.p. (°C) = 135–139. HRMS (ESI): Calcd. for C_31_H_23_N_4_O_5_ [M+H]^+^ 531.1668, found 531.1660.

2-((4-(((5-iodo-1,4-dioxo-1,4-dihydronaphthalen-2-yl)amino)methyl)-1*H*-1,2,3-triazol-1-yl)methyl)-2,3-dihydronaphtho[1,2-*b*]furan-4,5-dione (**20a**): The general procedure for triazole synthesis was followed by using 2-(azidomethyl)-2,3-dihydronaphtho[1,2-*b*]furan-4,5-dione (**7**) (51 mg, 0.2 mmol) and 5-iodo-2-(prop-2-yn-1-ylamino)naphthalene-1,4-dione (**18a**) (74 mg, 0.22 mmol) as the starting materials. Purification by column chromatography on silica gel (*n*-hexane/EtOAc 4:1) yielded **20a** (73 mg, 62%) as an orange solid. ^1^H NMR (600 MHz, DMSO-*d_6_*) *δ *= 8.33 (d, *J* = 2.8 Hz, 1H), 8.28 (d, *J* = 7.8 Hz, 1H), 8.10–8.08 (m, 1H), 7.99–7.88 (m, 2H), 7.83–7.75 (m, 1H), 7.61 (dd, *J* = 15.5, 7.2 Hz, 1H), 7.55–7.52 (m, 1H), 7.44–7.43 (m, 1H), 5.69–5.68 (m, 1H), 5.55–5.50 (m, 1H), 4.86 (dd, *J* = 14.8, 3.0 Hz, 1H), 4.76 (dd, *J* = 14.8, 7.3 Hz, 1H), 4.51–4.43 (m, 2H), 3.14 (dd, *J* = 15.4, 10.3 Hz, 1H), 2.80 (dd, *J* = 15.5, 6.6 Hz, 1H). ^13^C NMR (150 MHz, DMSO-*d_6_*) *δ *= 180.3 (C_q_), 179.8 (C_q_), 179.5 (C_q_), 174.9 (C_q_), 168.1 (C_q_), 148.8 (C_q_), 146.3 (CH), 143.4 (C_q_), 138.8 (CH), 135.0 (CH), 132.1 (CH), 129.5 (C_q_), 128.7 (CH), 126.8 (C_q_), 126.6 (CH), 124.4 (C_q_), 124.2 (CH), 115.0 (C_q_), 99.6 (CH), 94.2 (C_q_), 84.8 (CH), 79.3 (C_q_), 69.9 (CH_2_), 52.6 (CH_2_), 37.6 (CH_2_), 29.0 (CH). IR (KBr): *ṽ *= 3278, 2360, 1676, 1654, 1607, 1570, 1384, 1241, 658 cm^−1^. m.p. (°C) = 150–155. HRMS (ESI): Calcd. for C_26_H_18_IN_4_O_5_ [M+H]^+^ 593.0322, found 593.0320.

2-((4-(((5-hydroxy-1,4-dioxo-1,4-dihydronaphthalen-2-yl)amino)methyl)-1*H*-1,2,3-triazol-1-yl)methyl)-2,3-dihydronaphtho[1,2-*b*]furan-4,5-dione (**20b**): The general procedure for triazole synthesis was followed by using 2-(azidomethyl)-2,3-dihydronaphtho[1,2-*b*]furan-4,5-dione (**7**) (51 mg, 0.2 mmol) and 5-hydroxy-2-(prop-2-yn-1-ylamino)naphthalene-1,4-dione (**18b**) (50 mg, 0.22 mmol) as the starting materials. Purification by column chromatography on silica gel (*n*-hexane/EtOAc 4:1) yielded **20b** (63 mg, 65%) as an orange solid. ^1^H NMR (600 MHz, DMSO-*d_6_*) *δ *= 13.19 (s, 1H), 8.33 (t, *J* = 6.0 Hz, 1H), 8.09 (s, 1H), 7.76 (d, *J* = 7.8 Hz, 1H), 7.59 (dd, *J* = 17.4, 7.8 Hz, 2H), 7.52 (t, *J* = 7.2 Hz, 1H), 7.47 (dd, *J* = 13.2, 7.8 Hz, 2H), 7.27 (d, *J* = 7.8 Hz, 1H), 5.62 (s, 1H), 5.54 (d, *J* = 6.0 Hz, 1H), 4.86 (dd, *J* = 15.0, 2.4 Hz, 1H), 4.77 (dd, *J* = 15.0, 7.2 Hz, 1H), 4.53–4.45 (m, 2H), 3.15 (dd, *J* = 15.6, 10.2 Hz, 1H), 2.81 (dd, *J* = 15.6, 6.6 Hz, 1H). ^13^C NMR (150 MHz, DMSO-*d_6_*) *δ *= 188.1 (C_q_), 180.7 (C_q_), 180.3 (C_q_), 174.9 (C_q_), 168.0 (C_q_), 160.3 (C_q_), 149.4 (C_q_), 134.7 (CH), 134.5 (CH), 132.1 (CH), 130.5 (C_q_), 130.4 (C_q_), 128.7 (CH), 126.8 (C_q_), 125.3 (CH), 124.4 (CH), 124.1 (CH), 118.7 (CH), 115.0 (C_q_), 114.4 (C_q_), 99.1 (CH), 84.7 (CH), 52.6 (CH_2_), 40.1 (C_q_), 37.6 (CH_2_), 28.9 (CH_2_). IR (KBr): *ṽ *= 3383, 2361, 1681, 1616, 1572, 1469, 1384, 1255, 1230, cm^−1^. m.p. (°C) = 150–155. HRMS (ESI): Calcd. for C_26_H_19_N_4_O_6_ [M+H]^+^ 483.1305, found 483.1302.

2-((4-(((8-methoxy-1,4-dioxo-1,4-dihydronaphthalen-2-yl)amino)methyl)-1*H*-1,2,3-triazol-1-yl)methyl)-2,3-dihydronaphtho[1,2-*b*]furan-4,5-dione (**20c**): The general procedure for triazole synthesis was followed by using 2-(azidomethyl)-2,3-dihydronaphtho[1,2-*b*]furan-4,5-dione (**7**) (51 mg, 0.2 mmol) and 8-methoxy-2-(prop-2-yn-1-ylamino)naphthalene-1,4-dione (**18c**) (53 mg, 0.22 mmol) as the starting materials. Purification by column chromatography on silica gel (*n*-hexane/EtOAc 4:1) yielded **20c** (85 mg, 86%) as an orange solid. ^1^H NMR (600 MHz, DMSO-*d_6_*) *δ *= 8.67 (br s, 1H), 8.50 (br s, 1H), 8.26–8.25 (m, 1H), 8.21–8.19 (m, 1H), 8.06 (br s, 1H), 7.92–7.91 (m, 1H), 7.77–7.73 (m, 4H), 7.61 (br s, 1H), 5.89 (s, 1H), 4.65 (d, *J* = 11.8 Hz, 1H), 4.55 (br s, 2H), 4.31 (t, *J* = 10.3 Hz, 1H), 3.51 (s, 3H), 2.20 (d, *J* = 15.9 Hz, 1H), 2.09–2.05 (m, 1H). ^13^C NMR (150 MHz, DMSO-*d_6_*) *δ *= 181.1 (C_q_), 180.3 (C_q_), 179.4 (C_q_), 174.9 (C_q_), 168.0 (C_q_), 159.7 (C_q_), 149.2 (C_q_), 143.5 (C_q_), 136.2 (CH), 135.3 (C_q_), 134.8 (CH), 132.1 (CH), 130.4 (C_q_), 128.7 (CH), 126.8 (C_q_), 124.3 (CH), 124.2 (CH), 118.0 (CH), 117.0 (CH), 114.9 (C_q_), 98.8 (CH), 84.8 (CH), 56.5 (CH_3_), 52.6 (CH_2_), 40.1 (C_q_), 37.5 (CH_2_), 29.0 (CH_2_). IR (KBr): *ṽ *= 3430, 1612, 1572, 1384, 1284 cm^−1^.m.p. (°C) = 183–189. HRMS (ESI): Calcd. for C_27_H_21_N_4_O_6_ [M+H]^+^ 497.1461, found 497.1444.

2-((4-(((8-benzyloxy-1,4-dioxo-1,4-dihydronaphthalen-2-yl)amino)methyl)-1*H*-1,2,3-triazol-1-yl)methyl)-2,3-dihydronaphtho[1,2-*b*]furan-4,5-dione (**20d**): The general procedure for triazole synthesis was followed by using 2-(azidomethyl)-2,3-dihydronaphtho[1,2-*b*]furan-4,5-dione (**7**) (51 mg, 0.2 mmol) and 8-benzyloxy-2-(prop-2-yn-1-ylamino)naphthalene-1,4-dione (**18d**) (70 mg, 0.22 mmol) as the starting materials. Purification by column chromatography on silica gel (*n*-hexane/EtOAc 4:1) yielded **20d** (81 mg, 71%) as an orange solid. ^1^H NMR (600 MHz, DMSO-*d_6_*) *δ *= 8.08 (s, 1H), 7.96 (t, *J* = 6.0 Hz, 1H), 7.79 (d, *J* = 7.2 Hz, 1H), 7.76 (t, *J* = 8.4 Hz, 1H), 7.62 (d, *J* = 7.8 Hz, 2H), 7.59 (dd, *J* = 7.2, 0.6 Hz, 1H), 7.56 (d, *J* = 7.2 Hz, 1H), 7.52 (t, *J* = 6.6 Hz, 2H), 7.42 (t, *J* = 6.0 Hz, 3H), 7.34 (t, *J* = 7.8 Hz, 1H), 5.62 (s, 1H), 5.54–5.50 (m, 1H), 5.32–5.30 (m, 2H), 4.85 (dd, *J* = 15.0, 3.6 Hz, 1H), 4.76 (dd, *J* = 15.6, 7.8 Hz, 1H), 4.49–4.43 (m, 2H), 3.15 (dd, *J* = 15.0, 10.2 Hz, 1H), 2.80 (dd, *J* = 15.6, 6.6 Hz, 1H). ^13^C NMR (150 MHz, DMSO-*d_6_*) *δ *= 181.1 (C_q_), 180.3 (C_q_), 179.5 (C_q_), 174.9 (C_q_), 168.0 (C_q_), 158.6 (C_q_), 143.5 (C_q_), 136.8 (C_q_), 136.2 (CH), 135.3 (C_q_), 134.7 (CH), 132.1 (CH), 130.4 (C_q_), 128.7 (CH), 128.5 (CH), 127.8 (CH), 127.0 (CH), 126.8 (C_q_), 124.3 (CH), 124.1 (CH), 118.3 (CH), 118.2 (C_q_), 118.1 (CH), 115.0 (C_q_), 98.9 (CH), 84.8 (CH), 70.2 (CH_2_), 52.6 (CH_2_), 40.1 (C_q_), 37.5 (CH_2_), 29.0 (CH_2_). IR (KBr): *ṽ *= 3432, 1649, 1613, 1572, 1384, 1290, 1048 cm^−1^. m.p. (°C) = 115–120. HRMS (ESI): Calcd. for C_33_H_25_N_4_O_6_ [M+H]^+^ 573.1774, found 573.1764.

2-((4-(((1,4-dioxo-1,4-dihydroanthracen-2-yl)amino)methyl)-1*H*-1,2,3-triazol-1-yl)methyl)-2,3-dihydronaphtho[1,2-*b*]furan-4,5-dione (**20e**): The general procedure for triazole synthesis was followed by using 2-(azidomethyl)-2,3-dihydronaphtho[1,2-*b*]furan-4,5-dione (**7**) (51 mg, 0.2 mmol) and 2-(prop-2-yn-1-ylamino)anthracene-1,4-dione (**18e**) (58 mg, 0.22 mmol) as the starting materials. Purification by column chromatography on silica gel (*n*-hexane/EtOAc 4:1) yielded **20e** (65 mg, 63%) as an orange solid. ^1^H NMR (600 MHz, DMSO-*d_6_*) *δ *= 8.62 (t, *J* = 3.0 Hz, 1H), 8.47 (t, *J* = 2.4 Hz, 1H), 8.25 (d, *J* = 7.8 Hz, 1H), 8.20 (d, *J* = 7.2 Hz, 1H), 8.13 (s, 1H), 8.02 (t, *J* = 6.0 Hz, 1H), 7.78–7.71 (m, 3H), 7.64–7.63 (m, 1H), 7.52–7.49 (m, 1H), 7.47–7.45 (m, 1H), 5.85 (t, *J* = 1.8 Hz, 1H), 5.56–5.51 (m, 1H), 4.86 (dd, *J* = 15.0, 3.6 Hz, 1H), 4.77 (dd, *J* = 14.4, 7.2 Hz, 1H), 4.55–4.48 (m, 2H), 3.15 (dd, *J* = 15.6, 10.8 Hz, 1H), 2.81 (dd, *J* = 15.6, 6.6 Hz, 1H). ^13^C NMR (150 MHz, DMSO-*d_6_*) *δ *= 181.1 (C_q_), 181.0 (C_q_), 180.2 (C_q_), 174.8 (C_q_), 167.9 (C_q_), 149.1 (C_q_), 143.4 (C_q_), 135.1 (C_q_), 134.7 (CH), 133.6 (C_q_), 132.0 (CH), 130.2 (CH), 129.8 (CH), 129.8 (CH), 129.4 (C_q_), 128.9 (CH), 128.6 (CH), 128.5 (CH), 127.5 (C_q_), 126.7 (CH), 124.2 (CH), 124.1 (CH), 114.9 (C_q_), 102.4 (CH), 84.7 (CH), 54.9 (C_q_), 52.5 (CH_2_), 40.1 (C_q_), 37.4 (CH_2_), 29.0 (CH_2_). IR (KBr): *ṽ *= 3438, 2353, 1599, 1573, 1384, 1318, 1266 cm^−1^. m.p. (°C) = 158–163. HRMS (ESI): Calcd. for C_30_H_21_N_4_O_5_ [M+H]^+^ 517.1512, found 517.1499.

3-((4-(((5-iodo-1,4-dioxo-1,4-dihydronaphthalen-2-yl)amino)methyl)-1*H*-1,2,3-triazol-1-yl)methyl)-2,2-dimethyl-3,4-dihydro-2*H*-benzo[*h*]chromene-5,6-dione (**21a**): The general procedure for triazole synthesis was followed by using 3-(azidomethyl)-2,2-dimethyl-3,4-dihydro-2*H*-benzo[*h*]chromene-5,6-dione (**10**) (60 mg, 0.2 mmol) and 5-iodo-2-(prop-2-yn-1-ylamino)naphthalene-1,4-dione (**18a**) (74 mg, 0.22 mmol) as the starting materials. Purification by column chromatography on silica gel (*n*-hexane/EtOAc 4:1) yielded **21a** (81 mg, 64%) as an orange solid. ^1^H NMR (600 MHz, DMSO-*d_6_*) *δ *= 8.25 (d, *J* = 7.8 MHz, 1H), 8.15 (s, 1H), 8.08 (t, *J* = 6.0 MHz, 1H), 7.97 (d, *J* = 7.8 Hz, 1H), 7.86 (d, 7.2 Hz, 1H), 7.72 (d, *J* = 3.6 Hz, 2H), 7.58–7.55 (m, 1H), 7.41 (t, *J* = 7.8 Hz, 1H), 5.73 (s, 1H), 4.62 (dd, *J* = 13.8, 4.2 Hz, 1H), 4.49 (d, *J* = 6.0 Hz, 2H), 4.28 (dd, *J* = 13.8, 10.2 Hz, 1H), 2.46–2.41 (m, 1H), 2.17 (dd, 17.4, 5.4 Hz, 1H), 2.05 (dd, *J* = 17.4, 9.0 Hz, 1H), 1.50 (s, 3H), 1.34 (s, 3H). ^13^C NMR (150 MHz, DMSO-*d_6_*) *δ *= 179.9 (C_q_), 179.6 (C_q_), 178.9 (C_q_), 177.8 (C_q_), 160.2 (C_q_), 148.8 (C_q_), 146.2 (CH), 143.5 (C_q_), 135.9 (C_q_), 135.1 (CH), 134.9 (CH), 131.8 (C_q_), 131.1 (CH), 130.0 (C_q_), 129.6 (C_q_), 128.0 (CH), 126.5 (CH), 123.9 (CH), 123.9 (CH), 111.6 (C_q_), 99.8 (CH), 94.2 (C_q_), 80.7 (C_q_), 50.2 (CH_2_), 40.4 (CH), 37.7 (CH_2_), 26.6 (CH_3_), 21.4 (CH_3_), 19.9 (CH_2_). IR (KBr): *ṽ *= 3436, 2355, 2339, 1606, 1573, 1231, 729 cm^−1^. m.p. (°C) = 146–148. HRMS (ESI): Calcd. for C_29_H_24_IN_4_O_5_ [M+H]^+^ 635.0791, found 635.0792.

3-((4-(((5-hydroxy-1,4-dioxo-1,4-dihydronaphthalen-2-yl)amino)methyl)-1*H*-1,2,3-triazol-1-yl)methyl)-2,2-dimethyl-3,4-dihydro-2*H*-benzo[*h*]chromene-5,6-dione (**21b**): The general procedure for triazole synthesis was followed by using 3-(azidomethyl)-2,2-dimethyl-3,4-dihydro-2*H*-benzo[*h*]chromene-5,6-dione (**10**) (60 mg, 0.2 mmol) and 5-hydroxy-2-(prop-2-yn-1-ylamino)naphthalene-1,4-dione (**18b**) (50 mg, 0.22 mmol) as the starting materials. Purification by column chromatography on silica gel (*n*-hexane/EtOAc 4:1) yielded **21b** (60 mg, 57%) as an orange solid. ^1^H NMR (600 MHz, DMSO-*d_6_*) *δ *= 13.24 (d, *J* = 3.0 Hz, 1H), 8.37 (dd, *J* = 6.0, 4.2 Hz, 1H), 8.17 (d, *J* = 3.0 Hz, 1H), 7.89 (dd, *J* = 7.8, 4.2 Hz, 1H), 7.75 (t, *J* = 3.0 Hz, 2H), 7.60–7.54 (m, 2H), 7.50–7.48 (m, 1H), 7.25–7.23 (m, 1H), 5.71 (d, *J* = 1.8 Hz, 1H), 4.64 (dd, *J* = 13.8, 4.2 Hz, 1H), 4.53 (d, *J* = 6.6 Hz, 2H), 4.29 (dd, *J* = 13.2, 11.4 Hz, 1H), 2.48–2.43 (m, 1H), 2.18 (dd, *J* = 17.4, 5.4 Hz, 1H), 2.05 (dd, *J* = 17.4, 9.6 Hz, 1H), 1.51 (s, 3H), 1.36 (s, 3H). ^13^C NMR (150 MHz, DMSO-*d_6_*) *δ *= 188.2 (C_q_), 180.8 (C_q_), 178.9 (C_q_), 177.7 (C_q_), 160.3 (C_q_), 160.1 (C_q_), 149.5 (C_q_), 143.1 (C_q_), 135.0 (CH), 134.4 (CH), 131.7 (C_q_), 131.0 (CH), 130.0 (C_q_), 127.9 (CH), 125.3 (CH), 124.0 (CH), 123.9 (CH), 118.6 (CH), 114.4 (C_q_), 111.5 (C_q_), 99.3 (CH), 80.6 (C_q_), 50.1 (CH_2_), 40.3 (CH), 37.6 (CH_2_), 26.6 (CH_3_), 21.4 (CH_3_), 19.8 (CH_2_), 18.9 (C_q_). IR (ATR): *ṽ *= 3376, 2360, 1695, 1680, 1604, 1568, 1473, 1399, 1382, 1365, 1303, 1254, 1235, 1133, 842, 773 cm^−1^. m.p. (°C) = 187–189. HRMS (ESI): Calcd. for C_29_H_25_N_4_O_6_ [M+H]^+^ 525.1774, found 525.1756.

3-((4-(((8-methoxy-1,4-dioxo-1,4-dihydronaphthalen-2-yl)amino)methyl)-1*H*-1,2,3-triazol-1-yl)methyl)-2,2-dimethyl-3,4-dihydro-2*H*-benzo[*h*]chromene-5,6-dione (**21c**): The general procedure for triazole synthesis was followed by using 3-(azidomethyl)-2,2-dimethyl-3,4-dihydro-2*H*-benzo[*h*]chromene-5,6-dione (**10**) (60 mg, 0.2 mmol) and 8-methoxy-2-(prop-2-yn-1-ylamino)naphthalene-1,4-dione (**18c**) (53 mg, 0.22 mmol) as the starting materials. Purification by column chromatography on silica gel (*n*-hexane/EtOAc 4:1) yielded **21c** (42 mg, 39%) as an orange solid. ^1^H NMR (600 MHz, DMSO-*d_6_*) *δ *= 8.30 (s, 1H), 8.14 (s, 1H), 7.91 (d, *J* = 7.2 Hz, 1H), 7.81 (t, *J* = 6.0 Hz, 1H), 7.77–7.76 (m, 2H), 7.74 (d, *J* = 7.8 Hz, 1H), 7.62–7.59 (m, 1H), 7.55 (dd, *J* = 7.8, 0.6 Hz, 1H), 7.40 (d, *J* = 8.4 Hz, 1H), 5.65 (s, 1H), 4.63 (dd, *J* = 13.8, 4.8 Hz, 1H), 4.48 (d, *J* = 6.0 Hz, 2H), 4.28 (dd, *J* = 13.8, 10.2 Hz. 1H), 3.91 (s, 3H), 2.18 (dd, *J* = 17.4, 5.4 Hz, 1H), 2.06 (dd, *J* = 17.4, 9.0 Hz, 1H), 1.51 (s, 3H), 1.36 (s, 3H). ^13^C NMR (100 MHz, DMSO-*d_6_*) *δ *= 181.1 (C_q_), 179.4 (C_q_), 178.9 (C_q_), 177.7 (C_q_), 160.1 (C_q_), 159.7 (C_q_), 149.2 (C_q_), 143.5 (C_q_), 136.1 (CH), 135.3 (C_q_), 135.0 (CH), 131.7 (C_q_), 131.0 (CH), 130.0 (C_q_), 127.9 (CH), 123.8 (CH), 123.8 (CH), 117.9 (CH), 117.9 (C_q_), 116.9 (CH), 111.5 (C_q_), 98.9 (CH), 80.6 (C_q_), 56.4 (CH_3_), 50.0 (CH_2_), 40.3 (CH), 37.5 (CH_2_), 26.6 (CH_3_), 21.4 (CH_3_), 19.8 (CH_2_) cm^−1^. IR (KBr): *ṽ *= 3354, 2926, 2360, 1695, 1576, 1506, 1286, 1262, 1234, 1049. cm^−1^. m.p. (°C) = 140–142. HRMS (ESI): Calcd. for C_30_H_27_N_4_O_6_ [M+H]^+^ 539.1931, found 539.1956.

3-((4-(((8-benzyloxy-1,4-dioxo-1,4-dihydronaphthalen-2-yl)amino)methyl)-1*H*-1,2,3-triazol-1-yl)methyl)-2,2-dimethyl-3,4-dihydro-2*H*-benzo[*h*]chromene-5,6-dione (**21d**): The general procedure for triazole synthesis was followed by using 3-(azidomethyl)-2,2-dimethyl-3,4-dihydro-2*H*-benzo[*h*]chromene-5,6-dione (**10**) (60 mg, 0.2 mmol) and 8-benzyloxy-2-(prop-2-yn-1-ylamino)naphthalene-1,4-dione (**18d**) (70 mg, 0.22 mmol) as the starting materials. Purification by column chromatography on silica gel (*n*-hexane/EtOAc 4:1) yielded **21d** (65 mg, 53%) as an orange solid. ^1^H NMR (600 MHz, DMSO-*d_6_*) *δ *= 8.14 (s, 1H), 7.96 (t, *J* = 6.0 Hz, 1H), 7.89 (d, *J* = 7.8 Hz, 1H), 7.75–7.72 (m, 3H), 7.61 (d, *J* = 7.2 Hz, 2H), 7.59–7.58 (m, 1H), 7.56 (d, *J* = 7.2 Hz, 1H), 7.48 (d, *J* = 8.4 Hz, 1H), 7.41 (t, *J* = 7.8 Hz, 2H), 7.32 (t, *J* = 7.2 Hz, 1H), 5.67 (s, 1H), 5.29 (s, 2H), 4.63 (dd, *J* = 14.4, 4.8 Hz, 1H), 4.47 (d, *J* = 6.0, 2H), 4.29 (dd, *J* = 13.8, 9.6 Hz, 1H), 2.47–2.42 (m, 1H), 2.18 (dd, *J* = 17.4, 5.4 Hz, 1H), 2.06 (dd, *J* = 17.4, 9.0 Hz, 1H), 1.50 (s, 3H), 1.35 (s, 3H). ^13^C NMR (150 MHz, DMSO-*d_6_*) *δ *= 181.0 (C_q_), 179.6 (C_q_), 178.9 (C_q_), 177.7 (C_q_), 160.1 (C_q_), 158.5 (C_q_), 143.5 (C_q_), 136.7 (C_q_), 135.4 (C_q_), 135.0 (CH), 131.7 (C_q_), 131.0 (CH), 130.0 (C_q_), 128.4 (CH), 127.9 (CH), 127.7 (CH), 127.0 (CH), 123.8 (CH), 123.8 (CH), 118.2 (C_q_), 118.2 (CH), 111.5 (CH), 98.9 (CH), 80.6 (C_q_), 70.1 (CH_2_), 50.0 (CH_2_), 40.3 (CH), 37.5 (CH_2_), 31.4 (C_q_), 26.5 (CH_3_), 21.3 (CH_3_), 19.8 (CH), 18.9 (CH_2_), 13.9 (C_q_). IR (KBr): *ṽ *= 3384, 2924, 2853, 2363, 1609, 1573, 1286, 1260, 1232 cm^−1^. m.p. (°C) = 128–130. HRMS (ESI): Calcd. for C_36_H_31_N_4_O_6_ [M+H]^+^ 615.2244, found 615.2216.

3-((4-(((1,4-dioxo-1,4-dihydroanthracen-2-yl)amino)methyl)-1*H*-1,2,3-triazol-1-yl)methyl)-2,2-dimethyl-3,4-dihydro-2*H*-benzo[h]chromene-5,6-dione (**21e**): The general procedure for triazole synthesis was followed by using *N-*(1,4-dioxo-1,4-dihydronaphthalen-2-yl)acetamide (**10**) (43 mg, 0.2 mmol) and 2-(prop-2-yn-1-ylamino)anthracene-1,4-dione (**18e**) (58 mg, 0.22 mmol) as the starting materials. Purification by column chromatography on silica gel (*n*-hexane/EtOAc 4:1) yielded **21e** (66 mg, 59%) as an orange solid. ^1^H NMR (400 MHz, DMSO-*d_6_*) *δ *= 8.65 (s, 1H), 8.49 (s, 1H), 8.24 (d, *J* = 7.7 Mz, 1H), 8.19–8.18 (m, 2H), 8.06 (t, *J* = 6.3 Hz, 1H), 7.90 (d, *J* = 7.5 Hz, 1H), 7.76–7.68 (m, 4H), 7.62–7.58 (m, 1H), 5.88 (s, 1H), 4.64 (dd, *J* = 13.6, 4.4 Hz, 1H), 4.54 (d, *J* = 6.0 Hz, 2H), 4.30 (dd, *J* = 13.7, 10.2 Hz, 1H), 3.50 (s, 1H), 2.19 (dd, *J* = 17.5, 5.3 Hz, 1H), 2.06 (dd, *J* = 17.5, 9.3 Hz, 1H), 1.51 (s, 3H), 1.36 (s, 3H). ^13^C NMR (100 MHz, DMSO-*d_6_*) *δ *= 181.6 (C_q_), 181.4 (C_q_), 179.6 (C_q_), 178.0 (C_q_), 160.5 (C_q_), 149.6 (C_q_), 143.8 (C_q_), 135.4 (C_q_), 135.4 (CH), 134.0 (C_q_), 132.1 (C_q_), 131.4 (CH), 130.6 (CH), 130.4 (C_q_), 130.2 (CH), 130.2 (CH), 129.8 (C_q_), 129.3 (CH), 128.9 (CH), 128.3 (CH), 127.9 (C_q_), 127.1 (CH), 124.3 (CH), 124.2 (CH), 112.1 (C_q_), 103.1 (CH), 80.8 (C_q_), 70.2 (CH_2_), 50.4 (CH_2_), 40.7 (CH), 37.9 (CH_2_), 26.8 (CH_3_), 21.8 (CH_3_) cm^−1^. IR (KBr): *ṽ *= 3375, 2361, 1611, 1574, 1458, 1391, 1311, 1130 cm^−1^. m.p. (°C) = 227–231 (decomposition). HRMS (ESI): Calcd. for C_33_H_27_N_4_O_5_ [M+H]^+^ 559.1981, found 559.1956.

### 3.7. Anti-Tumor Assays

The in vitro cytotoxicity activity of the compounds was evaluated by the colorimetric MTT (3-(4,5-dimethylthiazol-2-yl)-2,5-diphenyl tetrazolium bromide)) assay [1] using the following tumor cell lines: HCT-116 (colon carcinoma), PC3 (prostate), SNB-19 (glioblastoma), K-562 (myelogenous leukaemia), HL-60 (human promyelocytic leukaemia), B16 (murine melanoma), A549 (human lung carcinoma), KG1 (human acute myeloid leukaemia), and RAJI (human Burkitt’s lymphoma), which were provided by the National Cancer Institute (Bethesda, MD, USA). The L929 cell line (mouse fibroblast L cells NCTC clone 929) employed in this study as a control cell line was obtained from the American Type Culture Collection (Manassas, VA, USA). The cell lines (Appendix A) were maintained in flasks containing RPMI 1640 or DMEM medium supplemented with 10% bovive fetal sérum, 100 U/mL penicillin, and 100 μg/mL streptomycin at 37 °C and in 5% CO_2_ atmosphere. The compounds tested were dissolved in DMSO. Doxorubicin served as the positive control. Cell treatments were performed with three replicates and the cells were mycoplasma free. After 72 h incubation, 100 μL of MTT solution (0.5 mg/mL) was added to each well, and the cells were incubated for 3 h. The supernatant was removed and 100 μL of DMSO was added and the absorbance at 595 nm was measured using Victor Nivo Multimode plate reader (PerkinElmer, Waltham, MA, USA). The absorbances obtained were used to calculate the IC_50_ values by nonlinear regression employing appropriate statistical software [56].

## 4. Conclusions

After the development of the research published by our research group in 2021 [32], it became clear that a sequel was necessary in order to fully evaluate and explore the anticancer activity that a synergetic combination of two naphthoquinoidal redox centres can offer. Through a copper catalysed 1,3-dipolar cycloaddition, fifteen new products were successfully achieved, each presenting astonishing anticancer activities against nine different cancer cell lines. Amongst those, the main activity was observed against the HL60 cell line, for which IC_50_ values as low as 0.3 μM were observed. This is a good result when compared to the positive control, doxorubicin, which has an IC_50_ value of 0.1 μM against the HL60 cell line. Along with these results, the cytotoxicity was also evaluated against the murine fibroblast cell line L929, in which it was possible to observe that compound **18b** is one the most selective ones, presenting IC_50_ values of 24 μM against the L929 and 1.8 μM against the HCT-116 cell line (selectivity index of 13.3), alongside compound **20e**, which presents an IC_50_ value of 15.9 μM against L929 and 1.6 μM against the HL60 cell line (selectivity index of 9.9). With these results in hand, the pathway towards a less aggressive additional therapy inches closer to reality, which may benefit thousands of people suffering from severe cases of cancer nowadays.

## Data Availability

Data is contained within the article and its respective Appendix A.

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
