# Peer review of "It Takes Two to Tango, Part II: Synthesis of A-Ring Functionalised Quinones Containing Two Redox-Active Centres with Antitumour Activities"

_molecules, 2023, doi:10.3390/molecules28052222_

Round 1
Reviewer 1 Report
Journal- Molecules
Manuscript ID: Molecules-2148563
Title-"It takes two to tango, part II: Synthesis of A-ring functionalised quinones containing two redox active centres with antitumor activities”
Reviewer comments
1. The introduction of the manuscript is not proper. It is short and does not properly define the current problem with existing anticancer drugs. Besides, the authors did not properly explain how quinone moiety-containing molecules can be a potential candidate as anticancer drugs. Authors need to rewrite the introduction section again and define each point properly.
2. Supplementary data for compound 9, six chemical shifts of protons associated with the structure are given, while the structure has only 4 secondary protons and one tertiary proton and 6 primary protons. In the text, one extra chemical shift of the proton is given. Authors need to justify it according to the structure.
3. Figure S1, what do you mean by at 50% probability level? Also, authors need to give the hydrogen bond distance formed between O3 and O2 of the quinone ring.
4. In the anticancer activity assay, for investigation of the anticancer effect, the author exposed each cell line for 72 h and achieved the highest potency against HL-60 cells. This is a very sensitive cell line, and upon prolonged incubation (72 h) cells may go under stress, and the IC50 may be falsely positive. In this case, authors can check the anticancer effect after 48 h of incubation and then go further in a time-dependent manner (http://dx.doi.org/10.1016/j.cbi.2017.07.001). Besides, before going to long incubation, authors need to doubling time of cell lines. So here the most important question is arising what is the criterion for selecting 72 of incubation?
Author Response
1. The introduction of the manuscript is not proper. It is short and does not properly define the current problem with existing anticancer drugs. Besides, the authors did not properly explain how quinone moiety-containing molecules can be a potential candidate as anticancer drugs. Authors need to rewrite the introduction section again and define each point properly.
Our response: We have edited and reorganized the introduction. Thanks for the suggestion.
2. Supplementary data for compound 9, six chemical shifts of protons associated with the structure are given, while the structure has only 4 secondary protons and one tertiary proton and 6 primary protons. In the text, one extra chemical shift of the proton is given. Authors need to justify it according to the structure.
Our response: We would like to thank the reviewer for the comment. We have reviewed the compound data and also checked the data previously described in the literature. The data are consistent. We also performed a complete review of the SI file to avoid any possible mistakes.
3. Figure S1, what do you mean by at 50% probability level? Also, authors need to give the hydrogen bond distance formed between O3 and O2 of the quinone ring.
Our response: Ellipsoids are geometrical objects drawn with six tensors (Uij), which are refined parameters to account discretely in a volume object the continuous electronic density of the atoms, integrating around 100% of the electron counts in the space delimited by the object. The ellipsoid probability (e.g., 50%) is therefore the volume scale of the object applied to structure picture clarity only. We have also inserted the data in the SI ‘distances in Angstroms: O3-H3 0.84(2); H3...O2 1.81(2); O2...O3 2.557(2); angle in degrees: O3-H3...O2 146.4(11)’.
4. In the anticancer activity assay, for investigation of the anticancer effect, the author exposed each cell line for 72 h and achieved the highest potency against HL-60 cells. This is a very sensitive cell line, and upon prolonged incubation (72 h) cells may go under stress, and the IC50 may be falsely positive. In this case, authors can check the anticancer effect after 48 h of incubation and then go further in a time-dependent manner (http://dx.doi.org/10.1016/j.cbi.2017.07.001). Besides, before going to long incubation, authors need to doubling time of cell lines. So here the most important question is arising what is the criterion for selecting 72 of incubation?
Our response: We agree that HL60 is a sensitive cell line. However, during our cytotoxicity assays we used a negative control composed of the cells in appropriate media and DMSO (less than 0.1%, as used in other treatments). No harmful effects on cell viability of the negative control were observed after 72 hours. Because many cytotoxic quinone compounds have effects on the cell cycle and HL60’s doubling time is about 36 to 48 hours, we ensured that the cells were treated for at least two cell cycles.
Author Response
1. The abstract is too short and needs to add more about quinones and cancer.
Our response: We would like to apologize for the mistake. The abstract has been improved and we believe it is now consistent with the work presented.
2. In the introduction, the authors directly jumped from cancer to bioactive molecules. There is a gap between cancer and bioactive molecules. It needs to add more information about the side effect of existing cancer therapies and why bioactive molecules are important as an alternative treatment for cancer.
https://www.mdpi.com/1422-0067/22/22/12455
https://www.eurekaselect.com/article/117007
Our response: We have edited and reorganized the introduction. We also cited the references suggested by the reviewer. Thanks for the excellent suggestion.
3. Authors wrote in line 37, “valuable bioactivities, including against Trypanosoma 38 cruzi [15], malaria [16], Aedes aegypti [17] and tuberculosis [18].” But not mentioned against various cancer, while quinones are well-known anticancer agents in different cancers.
Our response: We consider important to mention the potential of quinones and their various bioactivities, but we have revised the introduction to better fit it to the present work, related to antitumor activity.
4. Inconsistency in spacing …line 44, in HL60 cells [19].
Our response: It was corrected. Thank you!
5. The introduction is concise; it needs to add more information.
Our response: We have reformulated and improved the introduction. Thanks for the suggestion.
6. English language correction is needed to improve the quality of the manuscript.
7. Authors need to check typo errors throughout the manuscript.
Our response: We proofread the entire manuscript. Thanks for the suggestion.
Reviewer 3 Report
In this manuscript, Júnior and coworker reported a new series of bis-naphthoquinoidal derivatives and tested their anticancer activity against five different cancer cell lines. In a follow up study reported in ref 20 and ref 21, they combined two redox centers (naphthoquinoidal) using Cu-catalyzed click reaction. Tethering of two redox centers significantly improves the anticancer activity of the molecules and compounds 19a-19e have an impressive IC50 value of 0.3-1.1 μM. Considering the novelty and usefulness of these results, this referee recommends the manuscript for publication after addressing following comments.
1. IC50 for Doxorubicin against cancer cell line HL-60 (0.1 vs 0.02) and nontumor cell line L-929 (0.6 vs 1.72) significantly varied from the earlier data reported in ref 20 & ref 21. Which subsequently changed selectivity index of Doxorubicin against HL-60 to 6.0 from earlier reported value 86. Is there any particular reason behind this data variability? Author should address this issue.
2. Synergetic combination of two redox centers (naphthoquinoidal) significantly improved the anticancer activity compared to compounds with one redox center. Authors should provide a rationale behind this result.
3. In this work, para-quinones are considered as second redox center and combined with ortho quinones. Why authors didn’t consider another ortho-quinone moieties as second redox center and coupled with an ortho-quinone? (Ortho-quinone coupled ortho-quinones products)
4. Author should define A-ring and B-ring of the molecule in the general structure presented in Scheme 1, which will help reader to follow the manuscript.
5. Cytotoxic activity data of compound 7 against HL60 cell line was not determined. Is there any particular reason behind it? Otherwise, this data should be included in the revised version.
6. Concentration-response curve for all tested compounds (reported in Table 1) in antitumor assay should be included in the supporting information.
7. Line 25, 26: “nine different cancer cell lines” should be changed into “five different cancer cell lines”.
8. Line 199 and 200; azide “5” should be changed into azide “4”.
9. “Center” and “Centre” are interchangeably used in the manuscript. It should be consistent.
In Supporting information
10. 1H NMR spectrum for compound 19e, 20c, 21e are not clear and contain significant impurities. A clean NMR spectrum should be added to the revised version.

Author Response
1. IC50 for Doxorubicin against cancer cell line HL-60 (0.1 vs 0.02) and nontumor cell line L-929 (0.6 vs 1.72) significantly varied from the earlier data reported in ref 20 & ref 21. Which subsequently changed selectivity index of Doxorubicin against HL-60 to 6.0 from earlier reported value 86. Is there any particular reason behind this data variability? Author should address this issue.
Our response: Thanks for the checking the IC50 and selectivity index values from literature. We have revised all IC50 and selectivity index values in this work. We also inform that we are aware that many biological factors may influence in drug-dose response curves; therefore, cytotoxicity tests must be optimized according to cell type, passage number, and seeding density (https://doi.org/10.1038/s41598-020-62848-5). In this study, to improve replicability and generate reliable and reproducible IC50 data, we provided the seeding density for each cell in the article’s supplementary material. Also, the cell passage number was maintained at less than 18.
2. Synergetic combination of two redox centers (naphthoquinoidal) significantly improved the anticancer activity compared to compounds with one redox center. Authors should provide a rationale behind this result.
Our response: We have reorganized the introduction and included an explanation for the design of the new molecules. We have improved Scheme 1 for a better understanding of the strategy applied by us for the preparation of the novel compounds.
3. In this work, para-quinones are considered as second redox center and combined with ortho quinones. Why authors didn’t consider another ortho-quinone moieties as second redox center and coupled with an ortho-quinone? (Ortho-quinone coupled ortho-quinones products).
Our response: Thanks for the excellent suggestion. Actually, we are already working on the synthesis of new molecules considering ortho-quinone coupled ortho-quinones products. The results will be published in due course. Thank you.
4. Author should define A-ring and B-ring of the molecule in the general structure presented in Scheme 1, which will help reader to follow the manuscript.
Our response: It was done. Thank you!
5. Cytotoxic activity data of compound 7 against HL60 cell line was not determined. Is there any particular reason behind it? Otherwise, this data should be included in the revised version.
Our response: We had a small contamination problem with the respective strain and the data was not initially obtained. In this corrected version, we have included the missing data and also included antitumor activity data for 4 more tumor cell lines (B16, A549, KG1 and RAJI) evaluated with all compounds described in the paper.
6. Concentration-response curve for all tested compounds (reported in Table 1) in antitumor assay should be included in the supporting information.
Our response: All the concentration-response curves were inserted in the SI as requested.
7. Line 25, 26: “nine different cancer cell lines” should be changed into “five different cancer cell lines”.
Our response: Actually, we had started our studies with nine tumor cell lines, but unfortunately there was a deadline for submitting the manuscript and we were not able to complete all the analyses. We decided to submit the paper with 5 tumor cell lines, but we continued working to complete the results with the 9 cell lines that we had previously selected. In this updated and corrected version, we have included 4 more lineages, for a total of 9 tumor cell lines, so the data is correct in the article. Thanks for the comment and sorry for the initial mistake.
8. Line 199 and 200; azide “5” should be changed into azide “4”.
Our response: It was corrected. Thank you!
9. “Center” and “Centre” are interchangeably used in the manuscript. It should be consistent.
Our response: Sorry for the mistake. We have reviewed the overall manuscript again.
In Supporting information
10. 1H NMR spectrum for compound 19e, 20c, 21e are not clear and contain significant impurities. A clean NMR spectrum should be added to the revised version.
Our response: We include the spectra as requested by the reviewer and have reviewed the SI file.
Round 2
Reviewer 1 Report
Journal- Molecules
Manuscript ID: molecules-2148563
Title-" It takes two to tango, part II: Synthesis of A-ring functionalised quinones containing two redox active centres with antitumour activities”.
Reviewer comments
Dear Editor,
The authors have fulfilled all the queries/comments as it was asked previously. Now the manuscript is well written. I believe that it is an excellent piece of work for being published in the Molecules. Finally, I recommend that the paper be accepted for publication in its present form.
Decision- Accept
Reviewer 2 Report
Authors significantly improved the manuscript.